# Identification of motif-based interactions between SARS-CoV-2 protein domains and human peptide ligands pinpoint antiviral targets

Filip Mihalič [1], Caroline Benz [2], Eszter Kassa[2], Richard Lindqvist[3,4], Leandro Simonetti [2], Raviteja Inturi[1], Hanna Aronsson[1], Eva Andersson[1], Celestine N. Chi[1], Norman E. Davey[5], Anna K. Överby [3,4], Per Jemth [1] ✉ & Ylva Ivarsson [2] ✉

The virus life cycle depends on host-virus protein-protein interactions, which often involve a disordered protein region binding to a folded protein domain. Here, we used proteomic peptide phage display (ProP-PD) to identify peptides from the intrinsically disordered regions of the human proteome that bind to folded protein domains encoded by the SARS-CoV-2 genome. Eleven folded domains of SARS-CoV-2 proteins were found to bind 281 peptides from human proteins, and affinities of 31 interactions involving eight SARS-CoV-2 protein domains were determined ($K_D \sim$ 7-300 μM). Key specificity residues of the peptides were established for six of the interactions. Two of the peptides, binding Nsp9 and Nsp16, respectively, inhibited viral replication. Our findings demonstrate how high-throughput peptide binding screens simultaneously identify potential host-virus interactions and peptides with antiviral properties. Furthermore, the high number of low-affinity interactions suggest that overexpression of viral proteins during infection may perturb multiple cellular pathways.

The coronavirus disease 2019 (COVID-19) pandemic caused by the severe acute respiratory syndrome coronavirus 2 (SARS-CoV-2) virus overwhelmed hospitals around the world and pushed public health facilities to their limits in the past years. Rapid vaccine development has drastically reduced this burden, but given the high rate of vaccine breakthrough cases, better post-infection therapeutic interventions are needed. Currently, many academic and industrial laboratories are working to develop new drugs or testing the effect of repurposing existing drugs against SARS-CoV-2. For example, molnupiravir is a nucleoside analogue targeting viral RNA-dependent RNA polymerase[1],

and paxlovid is a combination of two protease inhibitors, one previously used against human immunodeficiency virus (HIV) and the other developed to target Nsp5 (the 3C-like protease or MPro) of coronaviruses[2]. However, to be prepared for future epidemics and pandemics, it is important to continue developing new broad-spectrum antivirals such that emerging viral threats can be treated directly and immediately with approved drugs, preferably administered orally. This is particularly important for SARS-like coronaviruses, given the efficiency of human-to-human transmission, and has been demonstrated by recurrent outbreaks: severe acute respiratory

[1]Department of Medical Biochemistry and Microbiology, Uppsala University, Box 582, Husargatan 3, 751 23 Uppsala, Sweden. [2]Department of Chemistry - BMC, Uppsala University, Box 576, Husargatan 3, 751 23 Uppsala, Sweden. [3]Department of Clinical Microbiology, Umeå University, 90185 Umeå, Sweden. [4]Laboratory for Molecular Infection Medicine Sweden (MIMS), Umeå University, 90187 Umeå, Sweden. [5]Division of Cancer Biology, The Institute of Cancer Research, 237 Fulham Road, London SW3 6JB, UK. ✉e-mail: Per.Jemth@imbim.uu.se; ylva.ivarsson@kemi.uu.se

syndrome coronavirus (SARS-CoV) in 2003, Middle East respiratory syndrome-related coronavirus (MERS-CoV) in 2012 and SARS-CoV-2 in 2019[3].

While most drugs work by inhibiting enzymes, the pharmaceutical industry is turning its attention to the more challenging but largely untouched area of protein-protein interaction interfaces. To this end, several approaches have been developed to identify novel drug targets by mapping virus-host protein–protein interactomes[4–7]. In one class of protein–protein interactions an intrinsically disordered region of a protein interacts with a folded domain of the binding partner. The most common interaction modules in disordered regions are short linear motifs (SLiMs). SLiMs are usually <10 amino acid residues long and can have degenerate sequences, with only 3–4 residues accounting for most of the specificity and affinity. Endogenous SLiM-based interactions have central roles in process such as cell signaling, regulation, cellular localizations and protein degradation[8]. Thus, by rewiring SLiM-based interactions of the host cell, viruses can exploit cellular systems and repurpose protein functions[9]. SLiMs can evolve ex nihilo through accumulation of one or a few mutations. Because viruses evolve relatively rapidly, viral mimicry of host protein SLiMs has been proposed as a commonly used strategy, and several examples have been described across most viral clades[6,7,10]. Indeed, proteins expressed from viral genomes often hijack the cellular machinery using SLiMs, which compete with native host protein-protein interactions. Conversely, folded domains of the viral proteome can interact with both human and viral SLiMs[9–13]. Each scenario provides a potential drug target, i.e., the folded domain of a human or viral protein can be targeted with a peptide or peptidomimetic.

The SARS-CoV-2 proteome consists of 29 proteins, encoded by 14 open reading frames (ORFs)[5]. The first two reading frames, ORF1a and ORF1b, encode 16 non-structural proteins (Nsp1 to Nsp16), formed after post-translational proteolytic cleavage by viral proteases. In addition, the SARS-CoV-2 genome encodes four structural proteins: spike (S), nucleocapsid (N), membrane (M), and envelope (E), as well as nine accessory proteins (ORFs 3a, 3b, 6, 7a, 7b, 8, 9b, 9c, and 10) (Fig. 1). The non-structural proteins facilitate viral mRNA replication and translation. Nsp1 is responsible for suppressing host translation while simultaneously promoting viral mRNA translation[14]. Nsp2 promotes viral evasion of the innate immune response[15]. Nsp3 serves as a central hub coordinating different steps of viral replication[16,17]. It consists of several stably folded globular domains, namely two ubiquitin-like domains (Ubl1, Ubl2), an ADP-ribose-1"-(phosphatase) hydrolase (ADRP), three SARS-unique domains (SUD-N, SUD-M, and SUD-C), a papain-like protease domain (PLpro) and a nucleic acid binding domain (NAB). Nsp3 is, together with Nsp4 and Nsp6 also involved in double membrane vesicle formation and organization[18,19]. Nsp5 serves as the main protease (MPro), that proteolytically processes the ORF1a and ORF1b into final proteins[20]. Nsp7, Nsp8, and Nsp12 form the core replication complex, with Nsp12 being the RNA dependent RNA polymerase (RdRp), and Nsp7 and Nsp8 forming a hexadecameric complex that enhances processivity[21–23]. In addition, Nsp9 also associates with the replication complex to promote 5'-capping of viral RNA[24]. The addition of a 5' cap to the viral RNA is crucial for RNA stability as well as efficient translation and requires several steps, with the Nsp10/Nsp14 and Nsp10/Nsp16 complexes also contributing to the final steps of the process[24]. In both cases, Nsp10 activates the catalytic activity of Nsp14 and Nsp16. Apart from its function in the 5'-capping process, Nsp14 also functions as an exoribonuclease facilitating the proofreading function of the replication complex. Finally, Nsp13 is a helicase, that unwinds the RNA and promotes efficient transcription[25], while Nsp15 serves as a uridine-specific endoribonuclease that facilitates the evasion of the host immune response[26]. Of the SARS-CoV-2 proteins, peptide binding has previously been shown for S receptor binding domain[27,28], Nsp3 Ubl1[11,29], Nsp5[30], and suggested for Nsp10[31], Nsp14[32], and Nsp16[33] based on computational analysis. Furthermore, a potential peptide binding site in Nsp9 has been suggested from structural analysis[34].

In the present study we systematically investigated the virus-human protein-protein interactome of folded protein domains encoded by the SARS-CoV-2 genome and peptides representing the intrinsically disordered regions of the human proteome. Several of the SARS-CoV-2 proteins contain multiple domains[35]. We focused on the folded domains rather than full length viral proteins to enable purification of bait proteins for phage display selections. Thus, SARS-CoV-2 protein domains were used as baits in proteomic peptide phage display (ProP-PD) screens against a peptide-phage display library that presents 1 million peptides from the intrinsically disorderd regions of the human proteome (called the Human Disorderome; HD2)[36]. The screen identified peptide-binding SARS-CoV-2 proteins and human peptide ligands that were validated with binding assays and tested as inhibitors in a SARS-CoV-2 infection assay. Our results suggest that multiple interactions with micromolar affinity may occur between protein domains of SARS-CoV-2 and human SLiMs. Some of these interactions may serve as targets for the design and development of peptidomimetic antivirals.

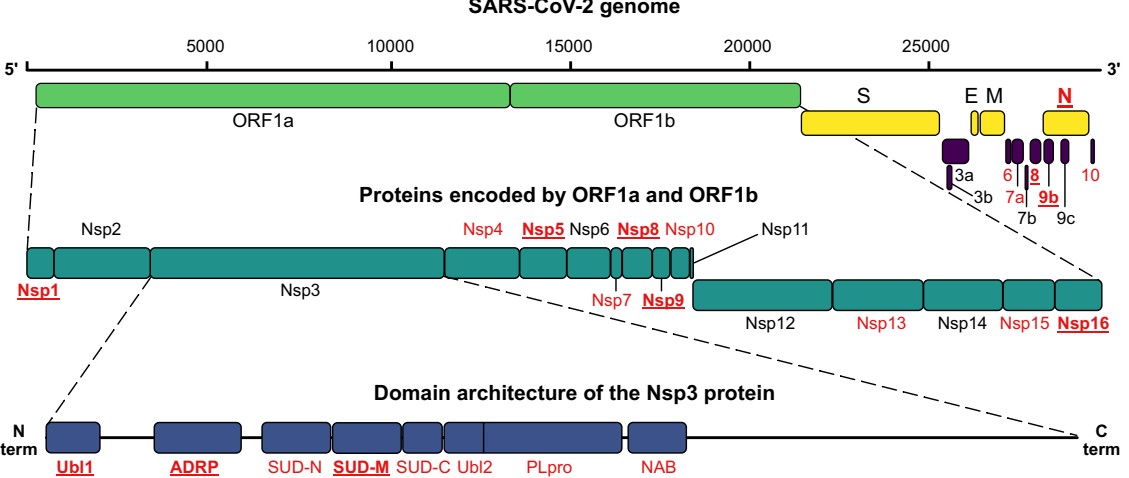

**Fig. 1 | Organization of the SARS-CoV-2 genome and proteome.** Proteins and protein domains that were successfully expressed in this study are shown in red. Proteins and protein domains that enriched ligands in ProP-PD selections are shown in bold and underlined. The figure is adapted from Gordon et. al.[5].

## Results

### Large-scale identification of human SLiM - SARS-CoV-2 protein interactions

We generated a collection of 31 expression constructs of known or predicted modular domains from 22 SARS-CoV-2 proteins, including two catalytically inactivated protease variants (Fig. 1; Supplementary Data 1). The domains were expressed as GST-tagged fusion proteins, and 26 of the 31 protein constructs were produced in sufficient quantities to be used as baits in phage display selections. The GST-fusion proteins were immobilized for triplicate selections in a 96-well plate and used in ProP-PD selections using our HD2 peptide phage-display library. This library displays 16 amino acid overlapping peptides that tile the intrinsically disordered regions of the human proteome on the surface of M13 phage[36]. The peptide-coding regions of the binding-enriched phage pools were analyzed by next-generation sequencing (NGS). Using previously established quality metrics (peptides found in replicate selections, overlapping peptides, high NGS score and/or motif-containing peptides) we found 281 high/medium (8/273) confidence peptides from 239 proteins interacting with 11 SARS-CoV-2 domains from nine viral proteins (Nsp1, Nsp3 (Ubl1, ADRP and SUD-M), Nsp5, Nsp8, Nsp9, Nsp16, Orf8, Orf9b and N NTD) (Fig. 1, Supplementary Data 2). The vast majority (118) of identified interactions involve Nsp9, followed by Nsp1 (47) and the catalytically inactivated Nsp5 (32). Based on the peptides identified by ProP-PD selections, consensus motifs could be established for two proteins using the SLiMFinder motif discovery tool[36,37] as implemented in PepTools[36] (Nsp5 [FLM][HQ][AS] and Nsp9 G[FL]xL[GDP]; Fig. 2). The Nsp9 binding motif is to our knowledge a novel motif. For Nsp5 (Mpro), the ligands may serve as potential substrates because the recognition motif resembles the preferred proteolytic site of the protease (LQ ↓ [GAS])[38].

A classical Gene Ontology (GO) term enrichment analysis was performed on the combined dataset of high/medium confidence interactions. The GO term enrichment analysis revealed enrichment of ligands associated with biological processes related to transcriptional regulation (Supplementary Data 3; *P*-values corrected for multiple testing using Benjamini–Hochberg correction <0.001). By comparing with the information available in protein interaction databases (collected August 2022, See Materials and Methods for details) we found that eighteen of the protein-protein interactions identified here were supported by reported interactions from previous studies (Fig. 2, Supplementary Data 2). The low overlap with other host-SARS-CoV-2 protein-protein interaction studies likely reflects technical differences between the approaches[39]. First, we used the folded parts of the proteins encoded by SARS-CoV-2 proteins in selections against the disordered regions of the human proteome. This limits the interactions that we can find to the peptide-mediated interactions involving these regions and domains. Second, the SLiM-based interactions are often of low affinity, and therefore underrepresented in AP-MS data[36]. Thus, it is expected to find a limited overlap between the interactomes generated by distinct methods. Finally, we note that the interactomes generated through large-scale studies on the SARS-CoV-2 host-virus interactomes have relatively low overlap even when using similar methods[40–42]. Importantly, we specifically searched for SLiM-based interactions of the viral proteins, which are often missed by other methods[36].

We selected 31 interactions for validation in a fluorescence polarization (FP) based binding assay, in which we first establish a valid fluorescein (FITC) labeled probe peptide, which was subsequently displaced in a second experiment with unlabeled peptide(s). The following SARS-CoV-2 protein domains were included in these binding experiments: Nsp1, Nsp3 Ubl1, Nsp3 ADRP, Nsp3 SUD-M, Nsp3 NAB, Nsp5, Nsp8, Nsp9, Nsp16, Orf9b, and N-NTD (Supplementary Data 4). Generally, peptides were selected to include highly enriched ligands (based on NGS counts) that were specific for the baits. For Nsp9 many

peptides were enriched, and we sampled ligands within a range of NGS counts. In addition, we included a set of peptides found with lower confidence that originated from proteins previously reported to interact with the bait protein (e.g., the PARP10 peptide binding to Nsp3 ADRP).

For Nsp3 NAB we did not detect binding with the probe peptides tested whereas weak, non-saturating binding was observed for Nsp1, Nsp8, Orf9b and N NTD, indicating that the interactions are of low affinity (Supplementary Fig. 1, Supplementary Data 4). In contrast, we conclusively confirmed the peptide binding capacity of six SARS-CoV-2 domains: Nsp3 Ubl1, Nsp3 ADRP, Nsp3 SUD-M, Nsp5, Nsp9, and Nsp16, as described below. While peptide binding has been suggested for several of the proteins, the peptide-binding of Nsp3 ADRP and Nsp3 SUD-M has to our knowledge not been previously reported.

### Characterization of SLiMs binding to Nsp3 domains

Coronavirus replication occurs primarily in double-membrane vesicles, derived from the host endoplasmic reticulum membrane, that provide protection from the host immune response[43]. Recently, the hexameric Nsp3 assembly was shown to form pores that span the double membrane and serve as a vital connection between the site of viral RNA synthesis inside the vesicle and the site of viral RNA translation in the cytoplasm[44]. Despite great recent efforts to map and characterize the SARS-CoV-2 interactome, the Nsp3 has been largely neglected due to its size and complexity, and therefore the functions of the various domains are poorly understood[5,42]. We expressed and purified all globular domains of Nsp3 and subjected them to ProP-PD selections (Fig. 1). Phage selections yielded a set of medium confidence ligands for Nsp3 Ubl1 (6 peptides), ADRP (3 peptides) and Nsp3 SUD-M (14 peptides) but without a clear consensus motif for any of the domains, in part due to the small number of retrieved peptides (Supplementary Data 2). We determined the equilibrium dissociation constant ($K_D$) using the FP assay for at least two human peptides for each of the three globular Nsp3 domains (Fig. 3A, Supplementary Fig. 1), which confirmed the binding of six of the tested peptides with affinities in the range of 20 to 300 μM (Supplementary Data 4). Nsp3 Ubl1 has previously been shown to bind to two peptide sequences in the disordered region of the N protein[11]. Our results reveal human ligands capable of binding Nsp3 Ubl1 and uncover that ADRP and SUD-M domains also have peptide binding capacities.

Three peptides derived from the protein NYNRIN (NYNRIN$_{1031-1046}$; CPSLSEEILRCLSLHD), the Nuclear receptor coactivator 2 (NCOA2) (NCOA2$_{1074-1089}$; SDEGALLDQLYLALRN), and from the Chromatin complexes subunit BAP18 (BAP18$_{40-55}$; AKWTE-TEIEMLRAAVK) were selected for affinity measurements with Nsp3 Ubl1. The measurements confirmed the binding of Nsp3 Ubl1 to NYNRIN$_{1031-1046}$ and NCOA2$_{1074-1089}$ (Fig. 3A, Supplementary Data 4), but not to BAP18$_{40-55}$. The two binding peptides shared some sequence similarity, and possessed a putative LxxLxL motif, known to adopt an alpha helical conformation upon binding[45,46]. One such interaction exists between the NCBD domain of CBP/p300 and the CID domains of the NCOA2 protein family[47]. We therefore tested the binding of Nsp3 Ubl1 to the entire CBP interaction domain (residues 1071–1110) of NCOA2 as well as to the two human paralogs NCOA1 and NCOA3 (residues 924-965 and 1045-1086 respectively). However, we did not detect any displacement for the two paralogues suggesting that the conserved LxxLxL motif is not the main driver of the interaction (Supplementary Fig. 1, Supplementary Data 4). To clarify which residues of the NCOA2 peptide are involved in the interaction, we performed a peptide SPOT array alanine scan, which revealed that the leucine residues in position 1 (P1), together with a tyrosine in position P5, was critical for binding (Fig. 3D), indicating that the Nsp3 Ubl1 binding motif in NCOA2$_{1074-1089}$ is LxxxY. This finding clarified the lack of binding of NCOA1 and NCOA3 paralogues in which valines are located at the corresponding tyrosine position. Similarly spaced

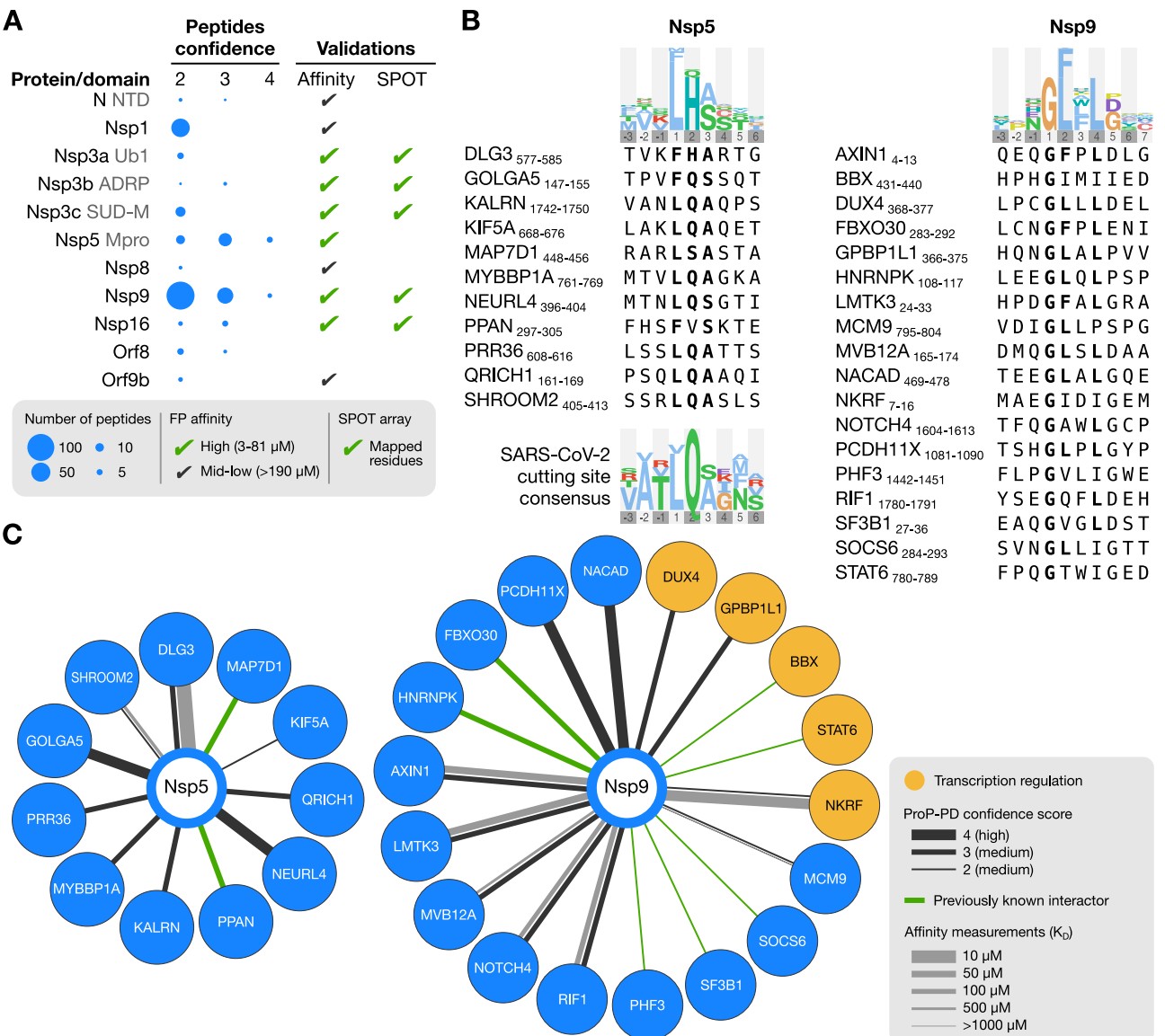

**Fig. 2 | Outline of the results. A** Overview of the results for the 11 SARS-CoV-2 protein domains that enriched peptides with medium and high confidence scores in ProP-PD selections. The number of peptides is proportional to the area of the blue circles. An overview of validations through fluorescence polarization-based affinity measurements and SPOT array is provided. **B** The motifs obtained from the selections with the SARS-CoV-2 proteins Nsp5 and Nsp9 are highlighted in bold in the alignments of the representative peptides. The consensus motif for the Nsp5 protease cleavage site is shown below the alignment[38]. For Nsp5, we chose to include peptides with an LQA motif matching the known proteolytic cleavage site

of the enzyme, explaining the apparent discrepancy between peptides and the consensus motif. The logos of the consensus motif were created using PepTools[36]. **C** Subset of identified interactions of SARS-CoV-2 Nsp5 and Nsp9. The thickness of dark lines shows the ProP-PD confidence scores for the interactions, and the thickness of gray lines shows the $K_D$ values obtained from the fluorescence polarization assays. Human proteins that contain the enriched GO terms associated with the transcriptional regulation (GO:0003700, GO:0000981, GO:0000976, GO:0001228, GO:0000978, GO:0000977, GO:0006355) are highlighted in orange.

hydrophobic residues are also found in the other Nsp3 Ubl1 binding peptides (Supplementary Fig. 2). To support the importance of the LxxxY motif in the NCOA2$_{1074-1089}$ peptide we tested a Y to V mutant of the peptide, which was found to bind with 20-fold lower affinity. We also introduced a tyrosine in the NCOA1$_{927-942}$ (V937Y), and NCOA3$_{1048-1063}$ (H1058Y) peptides and observed slightly higher affinities (1.5 to 3-fold) in binding experiments corroborating that the tyrosine contributes to the affinity (Supplementary Fig. 3, Supplementary Data 4). The lower affinity of the longer constructs suggest that the flanking regions of the motif also modulate affinity.

To test how the peptides might bind to Nsp3 Ubl1 we performed an in silico prediction using ColabFold, which is based on AlphaFold2[48]. Both the NCOA2$_{1074-1089}$ and the NYNRIN$_{1031-1046}$ peptides were predicted with a high per-residue confidence score (pLDDT > 80) to form

an alpha helix bound in the hydrophobic binding pocket between alpha helix 1 (α1) and helix 3 (α3) of Nsp3 Ubl1 (Fig. 3B; Supplementary Fig. 4).To validate this prediction, we mutated residues in the binding pocket (V852K, L889S, L893D; Supplementary Data 1) and measured binding to labeled NCOA2$_{1074-1089}$ peptide. As expected, the interaction was lost confirming that the peptides do in fact bind to the predicted binding pocket (Supplementary Fig. 1, Fig. 3E, Supplementary Data 4).

Intriguingly, a recently reported high-affinity interaction between the disordered region of the SARS-CoV-2 nucleoprotein (N; 219-LALLLL-224) and Nsp3 Ubl1 also engages the same hydrophobic binding pocket of Nsp3 Ubl1[11] (Supplementary Fig. 4A). In this case, the interaction is enhanced by an additional polar motif of N, that binds to Nsp3 Ubl1 in a distinct site. Therefore, to investigate if the ligands

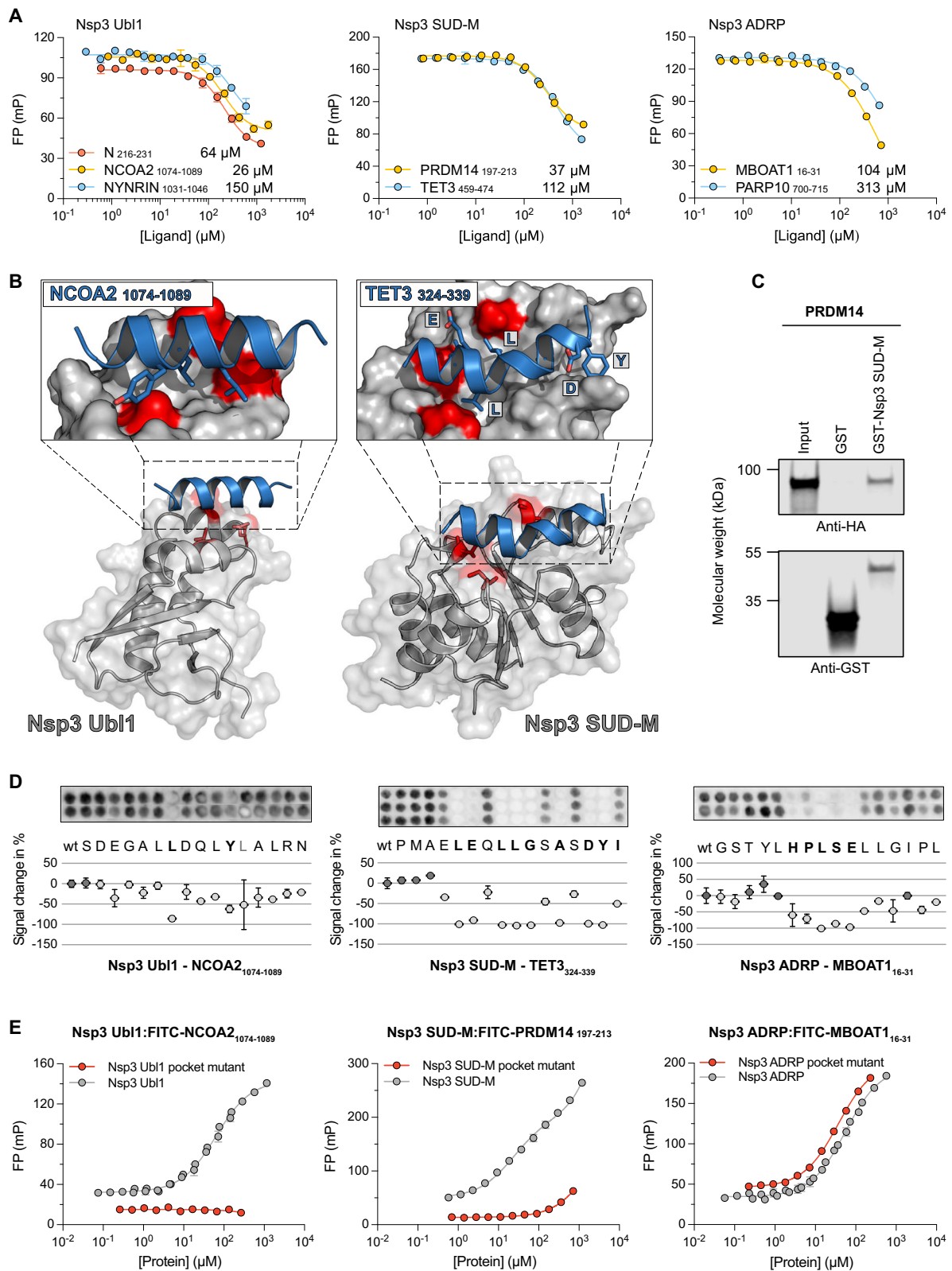

identified in this study directly compete with N for binding to Nsp3a Ubl1, we attempted to displace the FITC-labeled NCOA2$_{1074-1089}$ peptide probe from the Nsp3 Ubl1-probe complex with full length N. However, the FP signal increased upon the addition of N to the premixed Nsp3 Ubl1-probe sample as well as upon the addition of N to the sample containing only the probe peptide, rendering the displacement experiment inconclusive (Supplementary Fig. 1). Instead, we obtained

a N$_{216-231}$ peptide containing the LALLLL motif, and found it outcompete the FITC-NCOA2$_{1074-1089}$ peptide with comparable affinity (~2 fold higher $K_D$) to the NCOA2$_{1074-1089}$ peptide (Fig. 3A), thus supporting the notion that the two peptides bind to the same site.

Next, we focused on the binding of peptides to the other two Nsp3 domains, ADRP and SUD-M. The peptides that bind to Nsp3 ADRP were derived from Lysophospholipid acyltransferase 1 (MBOAT1$_{16-31}$;

**Fig. 3 | Biophysical analysis of the interactions between Nsp3 Ubl1, Nsp3 ADRP, and Nsp3 SUD-M, with peptide ligands from human proteins. A** Fluorescence polarization-monitored displacement experiments measuring the affinity between globular domains of Nsp3 and peptide ligands from disordered regions of the human proteome identified by phage display. The data are presented as means ± SD ($N$ = 3). The $K_D$ values for these and all subsequent affinity measurements performed in this study are shown next to each peptide and in Supplementary Data 4. **B** ColabFold structural predictions for the interaction between the globular domains of Nsp3 and the peptide ligands. The globular domains of Nsp3 are shown in gray, whereas the peptides are shown as blue ribbons. The residues that were mutated for the binding pocket analysis are highlighted in red (Ubl1: V852K, L889S, and L893D; SUD-M: A1397E, V1453K, S1478D) and the residues that were identified to be important for binding by SPOT arrays are shown as sticks. In the case of Nsp3 SUD-M the N terminus of the helix is on the left side of the enlarged panel. The

ColabFold pLDDT confidence scores were high for the globular domains of Nsp3 (>90) but varied widely for peptide predictions (>80 for interactions with Nsp3 Ub1 and 40–60 for Nsp3 SUD-M) and are shown in Supplementary Fig. 4. **C** Pulldown of full length PRDM14 by GST-tagged Nsp3 SUD-M as visualized by Western blot. Molecular weight is indicated. Original blots for these and all subsequent Western blot experiments are provided in the Source Data file (repeated in two independent experiments). **D** SPOT array alanine scans for the indicated peptides. Residues involved in binding are shown in bold. Signal intensities were normalized to wild type (wt) and presented as average percent signal change. Error bars indicate one standard deviation from the average (mean ± SD; $N \geq 2$). **E** Fluorescence polarization-monitored saturation experiments measuring the affinity between labeled peptides and globular domains of Nsp3 or the pocket mutant variants of them. The data are presented as means ± SD ($N$ = 6 for Nsp3 Ubl1 and Nsp3 ADRP measurements and $N$ = 3 for all other).

GSTYLHPLSELLGIPL) and Protein mono-ADP-ribosyltransferase PARP10 (PARP10$_{700-715}$; DGGTDGKAQLVVHSA), with the MBOAT1 peptide being the ligand with higher affinity (Fig. 3A, Supplementary Data 2 and Supplementary Data 4). The peptide SPOT array showed a strong signal for binding of the MBOAT1$_{16-31}$ peptide to Nsp3 ADRP, suggesting that each residue in the center of the peptide, 21-HPLSE-25, are critical for binding (Fig. 3D, residues in bold). A similar 4-YLSE-7 segment was also identified in AZIN2, an additional Nsp3 ADRP ligand found in the ProP-PD selection (Supplementary Fig. 2). The ColabFold predictions of the Nsp3 ADRP binding peptides did not converge with high confidence (pLDDT of ~30; Supplementary Fig. 4). Manual inspection indicated that the peptide binding was restricted to the surface between the N-terminal beta sheet (β1) and the C-terminal alfa helix (α6) (Supplementary Fig. 4). Thus, we introduced two mutations in Nsp3 ADRP (L1032R, L1191E) targeting the putative binding site. However, there was no effect on the affinity for the FITC-MBOAT1$_{16-31}$ peptide (Fig. 3E), therefore the peptide-binding site on Nsp3 ADRP remains to be established. Finally, we note that PARP10 is upregulated upon SARS-CoV-2 infection[49] and is known to mono-ADP-ribosylate amino acid residues as a part of host response to viral infections[50]. The Nsp3 SUD-N (also called Mac1) domain of SARS-CoV-2 is in turn a mono-ADP-ribosylhydrolase[51]. Thus, although the affinity of the Nsp3 ADRP-PARP10 interaction is relatively low ($K_D \approx 300$ μM), the interaction might be of relevance in the context of the virus counteracting the hosts response to viral infection.

The peptides used to study binding to Nsp3 SUD-M were from the PR domain zinc finger protein 14 (PRDM14$_{197-213}$; FTEEDLHFV-LYGVTPS) and Methylcytosine dioxygenase TET3 (TET3$_{324-339}$; PMAELEQLLGSASDYI). These were the most enriched peptides in the selection against Nsp3 SUD-M, and of these two, PRDM14$_{197-213}$ was the ligand with higher affinity (Fig. 3A, Supplementary Data 2 and Supplementary Data 4). The ColabFold analysis of Nsp3 SUD-M in complex with either PRDM14$_{197-213}$ or TET3$_{324-339}$ exhibited moderate pLDDT scores (between 50 and 70, Supplementary Fig. 4). While the two peptides did not share a specific binding motif, the SPOT array alanine scan showed distinct patterns of alternating amino acid residues interacting with Nsp3 SUD-M for both peptides (Fig. 3D, Supplementary Fig. 5). In the case of Nsp3 SUD-M: TET3$_{324-339}$ the ColabFold predictions and SPOT arrays converged allowing us to map a plausible binding interface, where TET3$_{324-339}$ forms an alpha helix exposing the underlined motif $_{324}$-PMAE**LEQ**LLG**SA**S**DY**I-$_{339}$ for interaction with a hydrophobic pocket in SUD-M (Fig. 3B, Supplementary Fig. 4C). We mutated the identified putative binding site in Nsp3 SUD-M (A1397E, V1453K, S1478D) and measured the affinity towards FITC-PRDM14$_{197-213}$ (Fig. 3E). The mutations reduced the affinity ~100 fold, thus supporting the notion that the peptide binds to the identified binding pocket. To further corroborate the interaction between Nsp3 SUD-M and human proteins, we expressed full length PRDM14 protein in HEK293T cells and performed GST-pulldown experiments using GST-tagged Nsp3 SUD-M. Full length PRDM14 was successfully

captured by the GST-Nsp3 SUD-M construct, but not by GST tag alone confirming that the SUD-M domain interacts with full length PRDM14 (Fig. 3C). Finally, we observed that the residues that were identified by SPOT arrays as crucial for binding to Nsp3 SUD-M are conserved, suggesting that they may be part of a motif involved in a native function (Supplementary Fig. 6).

Taken together we demonstrate that the Ubl1, ADRP, and SUD-M domains of Nsp3 possess the capacity to interact with human SLiMs. The identified peptides bound with intermediate micromolar affinity and the binding determinants of the peptides were defined by alanine scans. While the peptide-binding site on Nsp3 ADRP remains unknown, the peptide binding sites on Nsp3 Ubl1 and SUD-M were pinpointed using a combination of modeling, competitive peptide binding and mutagenesis.

## Identification of novel peptide ligands for Nsp5

Nsp5, or M$^{Pro}$, is the main protease, and a target of small molecule antiviral inhibitors such as the FDA-approved Paxlovid[2] as well as peptide inhibitors[30]. As Nsp5 recognizes and cleaves peptide ligands we used a catalytically dead variant as bait in the phage selection and identified a set of peptide ligands with a shared consensus motif ([FLM][HQ][AS]). We established the affinities of two peptides, the FHA containing DLG3$_{577-592}$ (TVKFHARTGMIESNR, $K_D$ = 7 μM), and the LQA containing SHROOM2$_{403-417}$ (GASSRLQASLSSSDV, $K_D$ = 280 μM) peptide (Fig. 4A). Using ColabFold we modeled the complex and found that the DLG3 peptide docks into the catalytic cleft of the protein (Fig. 4B, Supplementary Fig. 7). We therefore tested if the peptides could compete for substrate binding and thus act as inhibitors of catalytically active wild-type Nsp5. Indeed, both DLG3$_{577-592}$ and SHROOM2$_{403-417}$ worked as inhibitors in an assay with labeled peptide substrate (Fig. 4C). Our results expand the potential repertoire of peptide-based antiviral inhibitors against Nsp5[30,52–55].

## Peptides binding to Nsp9 share a consensus motif

Nsp9 from SARS-CoV-2 is a 113 amino acid residue long RNA-binding protein with a stable fold that shares 98% sequence identity with its SARS-CoV homolog[56,57]. Recent reports have established that Nsp9 is an essential component of the replication and transcription complex where it interacts with the Nidovirus RdRp-Associated Nucleotidyltransferase (NiRAN) domain of the RNA-dependent RNA polymerase Nsp12[58]. In this model, the monomeric Nsp9 binds to NiRAN via a C-terminal alpha helix and facilitates the addition of the GpppA-RNA cap to the 5′-end of the newly synthesized mRNA[24]. The same helix has been suggested to facilitate the formation of the Nsp9 homodimer with the crucial 100-GxxxG-104 motif at the dimer interface[34,59–62].

In our ProP-PD selections, we identified 147 human Nsp9 binding peptides in which a G[FL]xL[GDP] motif consensus was enriched (Fig. 2; Supplementary Data 2). We confirmed binding and determined the affinities for eight of the ligands, with $K_D$ values in the mid-micromolar range (Fig. 5A, B, Supplementary Fig. 1, Supplementary

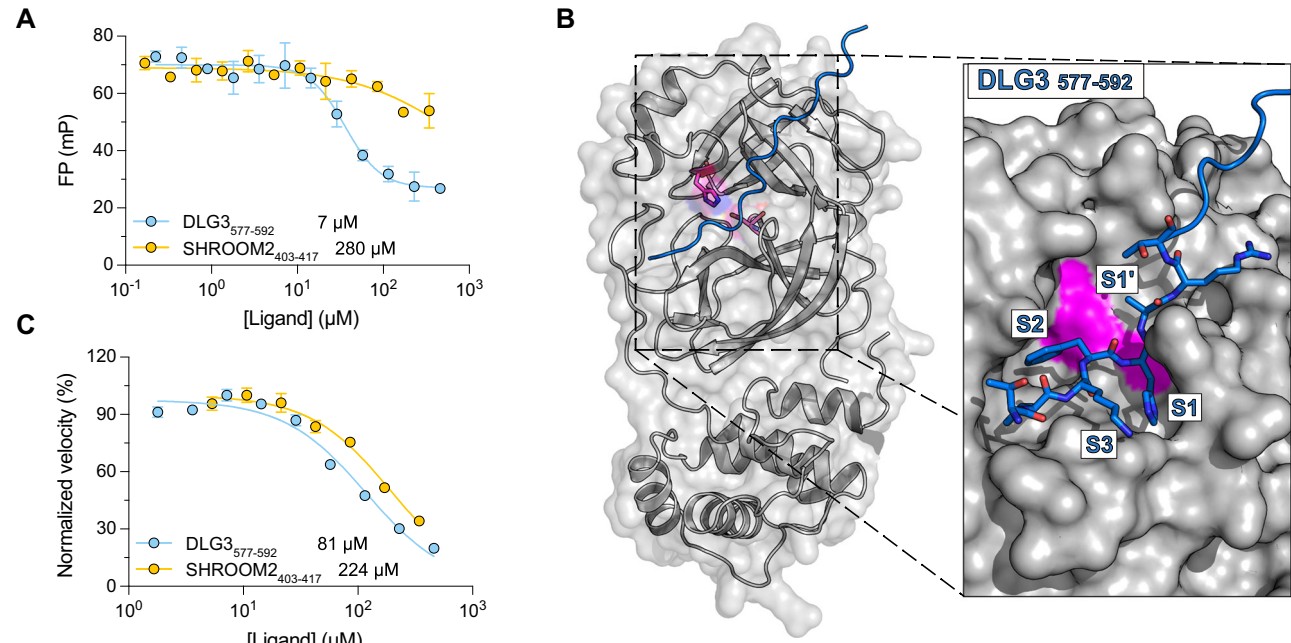

**Fig. 4 | DLG3 and SHROOM2-derived peptides bind to the catalytic pocket of Nsp5 and inhibit its protease activity. A** Fluorescence polarization-monitored displacement experiments measuring the affinity between Nsp5 and peptide ligands. Dissociation constants are indicated. The data are presented as means ± SD ($N = 3$). **B** ColabFold prediction of the interaction between DLG3$_{577-592}$ peptide and Nsp5. The catalytic residues Cys145 and His41 are shown in magenta. In the right panel the catalytic pocket is shown. The positions that correspond to the canonical substrate binding pockets are indicated and the interacting residues from

DLG3$_{577-592}$ are presented as sticks. S1', S1, S2, and S3 denotes residue position of the substrate peptide around the cleavage site. **C** Inhibition assay of Nsp5 catalytic activity. Reaction velocities were monitored at different concentrations of DLG3$_{577-592}$ and SHROOM2$_{403-417}$ peptides. IC$_{50}$ values are indicated next to the peptide name. Note that the peptides likely act as competing substrates rather than inhibitors. The initial reaction velocities were normalized to facilitate comparison. The data are presented as means ± SD ($N = 3$).

Data 4). The ligands with the highest affinity were peptides derived from the NF-kappa-B-repressing factor (NKRF$_{8-23}$; AEGIDI-GEMPSYDLVL), the protein kinase LMTK3 (LMTK3$_{22-36}$; PAHPDGFAL-GRAPLA), and the Axin-1 (AXIN1$_{1-15}$; MNIQEQGFPLDLGAS), which bound to Nsp9 with $K_D$ values of 30–50 μM (Supplementary Data 4). Of these, NKRF has been found to interact with other SARS-CoV-2 proteins including Nsp1[63] and Nsp10[64] with the latter interaction thought to regulate interleukin-8 production. A peptide from the Neurogenic locus notch homolog protein 4 (NOTCH4$_{1605-1620}$) displayed lower affinity, and bound to Nsp9 with a $K_D$ ~ 130 μM. Finally, we tested binding of a peptide corresponding to the C-terminus of Nsp9 (Nsp9$_{4237-4251}$; LNRGMVLGSLAATVR), as it has the GxxxG motif, but we did not detect any binding within the concentration range used in the competitive binding assay (Fig. 5A). To further dissect which residues of the peptides are essential for binding we performed a SPOT array alanine scan of the NKRF$_{8-23}$ and LMTK3$_{22-36}$ peptides. For the NKRF$_{8-23}$ peptide the analysis confirmed the consensus motif, with a substantial decrease in binding intensity when either of the two glycines at the positions P1 and P5 of the motif or the iso-leucine at position P4 were mutated to alanine. A minor effect was also observed upon mutation of the isoleucine at position P2 (Fig. 5C). Based on these results, we adjust the general motif to GΦxΦ[GDP], where Φ is a hydrophobic residue. However, for the higher affinity LMTK3$_{22-36}$ peptide the SPOT array only clearly confirmed the first glycine as a part of the motif. We further note that substitutions of residues following the consensus motif appears to enhance the affinity of the interaction, suggesting that the flanking residues may contribute to binding. The GΦxΦG motif in NKRF is conserved in jawed vertebrates and the GxxxG in LMTK3 is conserved in tetrapods, suggesting that the residues may potentially function as part of a native motif. However, in NOTCH4, the GxxxG is not conserved (Supplementary Fig. 6B).

In an attempt to define the binding pocket of the motif, we performed ColabFold prediction for peptide binding to either monomeric or dimeric Nsp9. This analysis however, showed low confidence scores (pLDDT <30; Supplementary Fig. 8) and did not converge on a similar binding mode for the three peptides. Because the peptide ligands have a GΦxΦ[GDP] motif, that is also found in the C-terminal helix of Nsp9 and facilitates dimerization, we hypothesized that Nsp9-binding peptides might interact with the dimerization interface and therefore interfere with the dimer formation[59,65]. To test whether the peptides bind via the C-terminal helix we expressed a Nsp9Δα construct, which is missing 20 residues of the C-terminus and a mutant where the two glycines in the C-terminal helix are substituted by larger polar and charged amino acids that should prevent dimer formation (G4240N, G4244D; Supplementary Data 1). While this Nsp9 double mutant bound to FITC-NKRF$_{8-23}$ with comparable affinities to the wild-type Nsp9, Nsp9Δα did not (Supplementary Fig. 1, Supplementary Fig. 9A, Supplementary Data 4), implying that binding of the peptides may involve the C-terminal helix but that the binding interface is not the GxxxG motif on the helix itself. This is also supported by the previous result showing that the Nsp9-derived peptide does not compete with the newly identified ligands (Fig. 5A). To confirm that the Nsp9Δα construct is folded we performed circular dichroism monitored temperature denaturation (Supplementary Fig. 9B).

To map the residues involved in the interaction between Nsp9 and the peptides, we recorded nuclear magnetic resonance (NMR) $^1$H$^{15}$N heteronuclear single quantum coherence (HSQC) spectra of $^{15}$N labeled Nsp9 at increasing concentrations of NKRF$_{8-23}$, LMTK3$_{22-36}$, and NOTCH4$_{1605-1620}$, respectively. Well-resolved spectra, which were in excellent agreement with the previously described NMR assignments[61,65], confirmed a folded protein and allowed us to monitor the changes in chemical shift perturbations upon addition of peptides (Fig. 5D, Supplementary Fig. 10, Supplementary Data 5). The

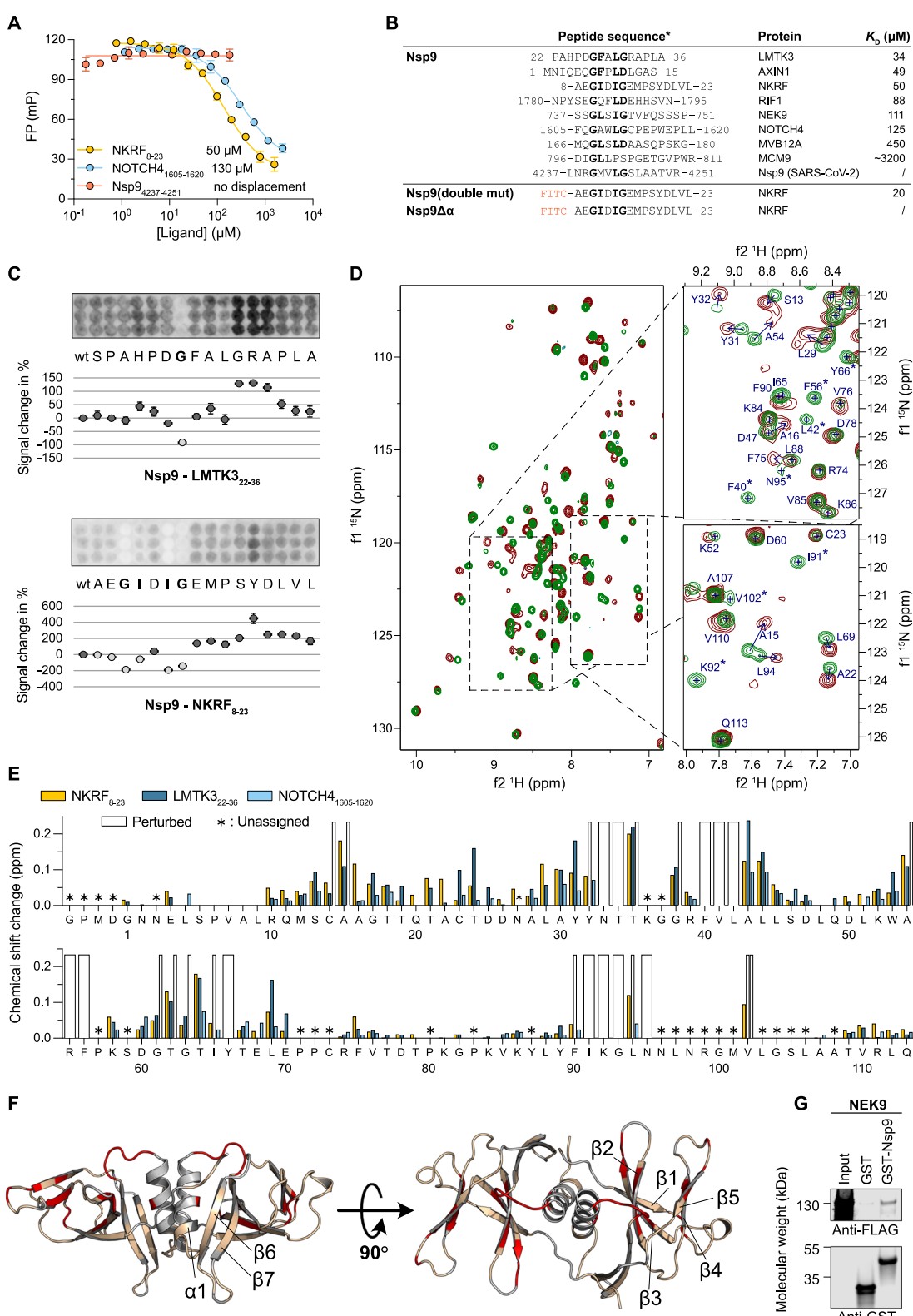

**B**

| Nsp9 | Peptide sequence* | Protein | $K_D$ (μM) |
|---|---|---|---|
| | 22-PAHPD**GFALG**RAPLA-36 | LMTK3 | 34 |
| | 1-MNIQEQ**GFPLD**LGAS-15 | AXIN1 | 49 |
| | 8-AE**GIDIG**EMPSYDLVL-23 | NKRF | 50 |
| | 1780-NPYSEGQF**LD**EHHSVN-1795 | RIF1 | 88 |
| | 737-SS**GLSIG**TVFQSSSP-751 | NEK9 | 111 |
| | 1605-FQ**GAWLG**CPEPWEPLL-1620 | NOTCH4 | 125 |
| | 166-MQ**GLSLD**AASQPSKG-180 | MVB12A | 450 |
| | 796-DI**GLL**PSPGETGVPWR-811 | MCM9 | ~3200 |
| | 4237-LNRG**MVLG**SLAATVR-4251 | Nsp9 (SARS-CoV-2) | / |
| Nsp9(double mut) | FITC-AE**GIDIG**EMPSYDLVL-23 | NKRF | 20 |
| Nsp9Δα | FITC-AE**GIDIG**EMPSYDLVL-23 | NKRF | / |

substantial overlap of chemical shift perturbation changes upon binding of the three peptides suggested an overall conserved binding interface (Fig. 5E). We focused on the residues that exhibited chemical shift changes of at least one standard deviation above average for all three peptides. The data suggested significant rearrangements of residues in the hydrophobic core of Nsp9 in the β1–β5 region while β6 and β7 remained unperturbed (Fig. 5F, Supplementary Fig. 10H–I). Moreover, comparing the surface exposed residues that are

perturbed showed enrichment of affected residues in the area formed by the loop between β sheets 2 and 3, loop between β sheets 4 and 5, and the loop that is N-terminal of the C-terminal helix (Supplementary Fig. 11). Importantly, chemical shifts for the C-terminal alpha helix were not observed as discussed previously[61,65], leaving open the possibility that the peptide-binding could involve the helix without showing any perturbations in the NMR HSQC spectra. Interestingly a previous study has observed that short peptides

**Fig. 5 | Interactions between Nsp9 and human peptide ligands are mediated by a GΦxΦ[GDP] motif and lead to rearrangement of the hydrophobic core of Nsp9. A** Representative FP-monitored displacement experiments measuring the affinity between Nsp9, two human peptide ligands and a peptide from Nsp9 with the motif. Affinities are shown next to each peptide. The data are presented as means ± SD. ($N = 3$) **B** Alignment of peptides identified by ProP-PD for which the affinities were measured. Residues, corresponding to the identified GΦxΦ[GDP] motif are shown in bold. For Nsp9Δα we did not observe saturation with the FITC-NKRF$_{8-23}$ peptide. **C** SPOT array alanine scans for the indicated peptides. Signal intensities were normalized to wild type (wt) and presented as percentage signal change (mean ± SD, $N = 3$). **D** HSQC experiments showing the $^1$H-$^{15}$N peak shifts. The left panel shows the superposition of Nsp9 protein (green) and Nsp9 mixed with NKRF peptide (wine red). The right panels show representative examples of the observed chemical peak shift perturbations. The arrows indicate the directions of the chemical shift change, and the asterisk indicates that the peaks disappeared

after addition of the peptide, indicating a large perturbation of the chemical environment. The full spectra are shown in Supplementary Fig. 10. Numbering is according to the start of Nsp9 after proteolytic processing. **E** Chemical shift changes of each residue upon the addition of peptide ligands. Perturbed means that the peak disappeared upon the addition of the ligand indicating large effect. Asterisk denotes residues which could not be unambiguously assigned. **F** Representation of the residues which displayed large change in chemical shift after addition of the NKRF$_{8-23}$ peptide, as observed by NMR experiments. Residues that changed more than one standard deviation above the average are colored in red, the residues below this threshold are beige, and the residues whose peak shifts could not be unambiguously assigned are gray. The PDBid: 6wxd model of Nsp9 dimer was used for visualization. **G** The interaction between the full length NEK9 and GST-tagged Nsp9 by pull-down experiments, visualized by western blot. GST-tag alone was used as negative control (two independent experiments).

co-crystalized with Nsp9 forming contacts in the same area as the peptides identified in this study[34].

To determine whether the Nsp9 is a dimer in solution, and if peptide binding interferes with dimer formation, we measured the $T_1$ and $T_2$ relaxation times, which report on global motion of individual residues of the protein (Supplementary Data 6). The ratio of the two relaxation times, gives the rotational correlation time $\tau_c$, which was 15.3 ns for the free Nsp9 protein and 9.9 ns for the protein in complex with NOTCH4$_{1605-1620}$ peptide corresponding to estimated molecular weights of 29 and 18.8 kDa respectively. Because the molecular weight calculation from $\tau_c$ is highly dependent on the shape of the molecule, the experiments only provide a rough estimate of the size. Thus, while the predicted molecular weight for a Nsp9 homodimer (25.7 kDa) and a Nsp9 monomer bound to the NOTCH4 peptide (14.7 kDa) correlates well with dimer and monomer, respectively, these results should be treated with caution as several reports suggested that the dimer formation occurs only at high micromolar concentrations of Nsp9[59,65].

Finally, to determine if Nsp9 interacts with identified proteins also in the context of full-length proteins we expressed NEK9 and LMTK3 in HEK293T cells and attempted pulldowns with GST tagged Nsp9. We successfully confirmed the interaction of Nsp9 with full length NEK9 (Fig. 5G), but not with full-length LMTK3 probably due to low expression levels (Supplementary Fig. 12).

In summary, we identified short peptide binders that interact with Nsp9 via a GΦxΦ[GDP] motif. This motif is present in many human proteins (Figs. 2B, 5B), but there is to our knowledge no reported binder of such a motif in the human proteome. Nevertheless, the interaction between Nsp9 and the human proteins could affect viral replication or disturb normal cell function.

## Peptides directly interfere with the Nsp10-Nsp16 complex

Since the SARS-CoV-2 virus does not have access to the cellular mRNA capping machinery, its genome encodes a full set of enzymes that facilitate complete 5′ capping of viral RNA, thereby promoting translation and evasion of the immune response[24,66,67]. In the final step of this process the methyltransferase Nsp16 catalyzes the C2′-O methylation of the first nucleotide of the RNA, forming a fully functional $^{7Me}$GpppA$_{2'-O-Me}$-RNA[24,68,69]. To catalyse this reaction Nsp16 forms a complex with Nsp10, which in turn activates Nsp16 by stabilizing the vital S-adenosylmethionine binding pocket[66,68].

While ProP-PD selections against Nsp10 did not result in specific enrichment of peptides, we identified peptides from six proteins that bind to Nsp16 (Supplementary Data 2, Supplementary Fig. 2) of which we selected three peptides for affinity measurements. These peptides were derived from Dual specificity tyrosine-phosphorylation-regulated kinase 1B (DYRK1B$_{395-410}$; PGHSPADYLRFQDLVL), Islet cell autoantigen 1-like protein (ICA1L$_{450-465}$; NQDMSAWFNLFADLDP), and uncharacterized protein FLJ43738 (FLJ43738$_{76-91}$; EDPLDSYLNFQALISP). Conservation analysis showed that the peptide region from DYRK1B is

conserved among jawed vertebrates, and the ICA1L peptide and FLJ43738 peptides are conserved mainly in mammals (Supplementary Fig. 6). Using the FP assay, we confirmed the interactions and determined that the peptides interacted with Nsp16 with $K_D$ values of 56, 30 and 62 μM, respectively (Fig. 6A). Interestingly, the tightest binder, FITC-labeled DYRK1B (FITC-DYRK1B$_{394-409}$), bound with a $K_D$ of ~3 μM, exhibiting a 10-fold higher affinity compared to the unlabeled peptide with the same sequence (DYRK1B$_{394-409}$) (Supplementary Fig. 1, Supplementary Data 4) implying that the FITC label itself contributes to binding. Alanine scanning SPOT array analysis of the ICA1L$_{450-465}$ and DYRK1B$_{395-410}$ peptides revealed the presence of a hydrophobic motif (WxxxF) in the ICA1L$_{450-465}$ peptide, which is also present in the peptide from the homologous ICA1 (Fig. 6B, Supplementary Fig. 2). For the DYRK1B$_{395-410}$ peptide the SPOT array analysis established the importance of a phenylalanine in P1 but did not confirm the second position of the hydrophobic motif (P5, which in this case would be a valine (Fig. 6B, Supplementary Fig. 2). A putative [WF]xxxΦ motif is also present in the validated binding peptide from FLJ43738$_{76-91}$ (Supplementary Fig. 2). ColabFold predicted that the three validated peptides form an alpha helix and bind to the hydrophobic groove formed by alpha helices α3, α4 and α10 of Nsp16 (Fig. 6C, Supplementary Fig. 12). This result correlated with the SPOT array data and corroborated the importance of hydrophobic residues at the P1 and P5 positions of the Nsp16 binding motif.

Since the same hydrophobic interface of Nsp16 facilitates the interaction with Nsp10[69] we hypothesized that our newly characterized peptide ligands could compete with Nsp10 for binding to Nsp16 (Fig. 6C). Indeed, we demonstrated that full-length Nsp10 directly competes with the labeled FITC-DYRK1B$_{394-409}$ peptide in a displacement experiment (Fig. 6D). This experiment also allowed us to determine the $K_D$ of the interaction between Nsp10 and Nsp16 to be 5.5 μM.

Lastly, we attempted capture of full length ICA1L protein by GST-tagged Nsp16 in a GST-pulldown experiment, but GST-tagged Nsp16 failed to co-precipitate ICA1L (Supplementary Fig. 13). The ColabFold prediction of full ICA1L predicted residues 450–465 to have helical propensity but not to be part of the folded domain (Supplementary Fig. 14). However, adjacent folded regions of the motif could interfere with binding between ICA1L and Nsp16.

Overall, we identified several peptide binders that interact with Nsp16, measured their affinity, and determined their binding interface with Nsp16. The peptide ligands compete with Nsp10 for binding to Nsp16 and could therefore act as inhibitors of the essential Nsp10-Nsp16 interaction, and hamper viral replication.

## Inhibition of viral proliferation

We selected eleven of the peptide ligands for further experiments in virus infection assays. We designed lentiviral expression constructs, expressing four repeats of each peptide interspaced by flexible Gly-SerThr linkers and conjugated C-terminally to an enhanced green

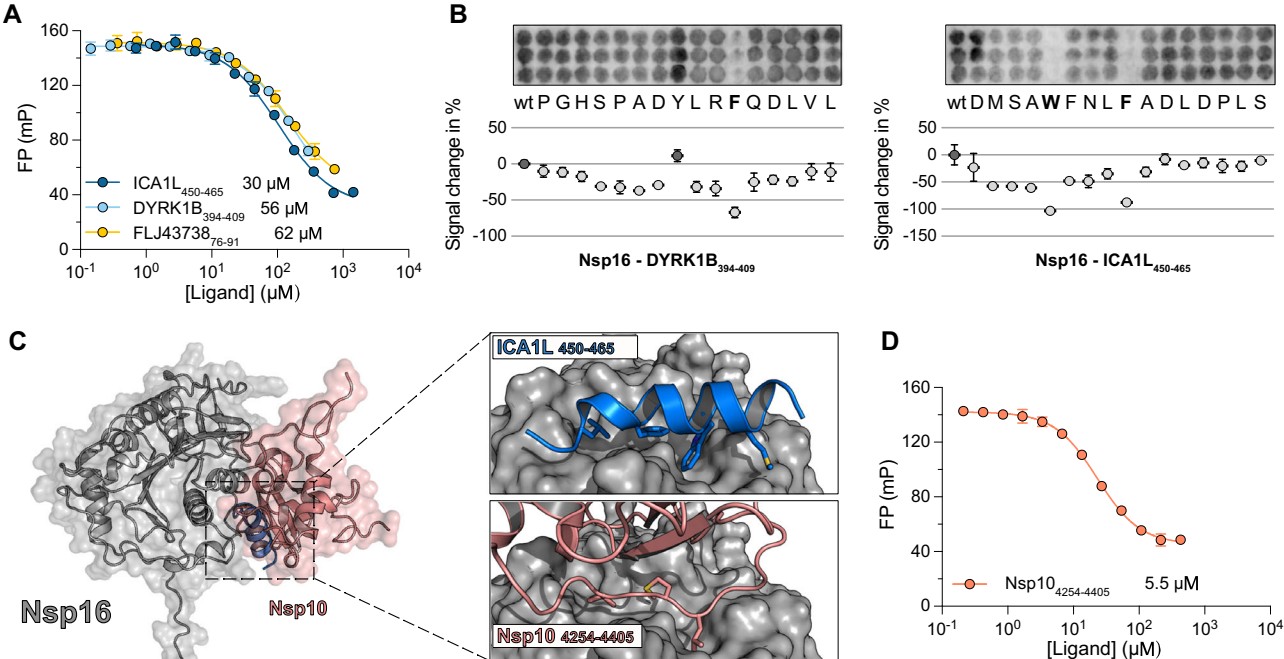

**Fig. 6 | Peptides identified with ProP-PD directly compete with Nsp10 for binding to Nsp16. A** Displacement experiments measuring affinity between Nsp16, and three peptide ligands derived from human proteins. $K_D$ values are shown next to each peptide. The data are presented as means ± SD ($N = 3$). **B** SPOT array alanine scans for the DYRK1B$_{395-410}$ and ICA1L$_{450-465}$ peptides. Residues involved in binding are shown in bold. Signal intensities were normalized to wild type (wt) and presented as percent signal change (mean ± SD, $N = 3$). **C** ColabFold structural predictions for the interaction between Nsp16 (gray), and the ICA1L$_{450-465}$ peptide (blue). The predicted structure is superimposed onto the crystal structure of the Nsp16-Nsp10 complex (PDBid: 7jyy). The right panels show a magnified view of the peptide binding pocket highlighting the similarities of the Nsp16 binding to the peptide or Nsp10, respectively. The pIDDT score for the ICA1L$_{450-465}$ peptide was ~60 (see Supplementary Fig. 13). **D** FP-monitored displacement by Nsp10 of a complex between Nsp16 and FITC-DYRK1B$_{394-409}$ shows that Nsp10 and the peptide binds to the same surface of Nsp16. The $K_D$ value for the Nsp10-Nsp16 interaction is indicated. The data are presented as means ± SD ($N = 3$).

fluorescent protein (EGFP) (Supplementary Data 7). VeroE6 cells were first transduced with the lentiviruses and, after 72 h, infected with SARS-CoV-2 for 16 h, after which the number of infected cells was determined. We found that five of the lentiviral constructs reduced the production of viral particles, namely the Nsp3 ADRP binding EGFP-PARP10 construct, the Nsp9 binding EGFP-NOTCH4 and EGFP-LMTK3 constructs, and the Nsp16 binding EGFP-DYRK1B and EGFP-ICA1L constructs (Fig. 7A, B). The low affinity peptide ligands binding to N-NTD and Orf9b, and the NEK9-derived ligand binding to Nsp9 showed instead pro-viral effects when introduced in the cell, which in the two first cases likely depend on off-target effects. It could also be speculated that the effect observed for the low-affinity Orf9b ligand could affect the dimeric state of the protein, or its fold-switching interaction with Tom70[70]. The pro-viral effect of the Nsp9-binding ligand is more difficult to explain, given that the other two Nsp9 ligands tested had antiviral effects. Nevertheless, we confirmed the interference of the ligands with viral replication by treating the infected cells with the wild-type NOTCH4 and DYRK1B-derived peptides, or negative control peptides with mutated binding motifs, fused to the cell-penetrating HIV Tat-derived sequence (YGRKKRRQRRRGSG) (Supplementary Data 8). These experiments showed inhibitory effects of the Nsp9-binding wild-type NOTCH4$_{1605-1620}$, as well as the Nsp16 binding DYRK1B$_{394-409}$ peptides, and less inhibition by the negative controls (Fig. 7C, D), establishing the ligands as potential starting points for the development of peptidomimetic inhibitors of SARS-CoV-2 infection. To investigate if the peptides specifically inhibit SARS-CoV-2 replication, we also treated human coronavirus 229E (HCoV-229E) infected MRC5 cells with the cell penetrating constructs. Interestingly, the peptides failed to inhibit HCoV-229E infection (Supplementary Fig. 15), suggesting a beta-coronavirus specific inhibitory effect. We noticed that the DYRK1-

derived peptide was less efficient inhibitor of viral replication when presented as a Tat-tagged version in comparison to the GFP-tagged version produced intracellularly. We therefore investigated the cellular uptake of the Tat-tagged peptides and found that while the Tat-NOTCH4 peptide was readily internalized, the uptake of the Tat-tagged-DYRK1B peptide was much less efficient, potentially explaining the low efficacy of the peptide (Fig. 7E). Finally, we reasoned that if the NOTCH4-derived peptide act by targeting Nsp9 then it should block RNA replication. We therefore conducted a time of addition experiment where we added the inhibitor at distinct time points (1, 3, or 5 h post infection; Fig. 7F). The results showed that the inhibitor has the most potent effect 3 h post infection, supporting the notion that it blocks RNA replication rather than interfering with viral entry or egress.

## Discussion

In the present work, we performed a large-scale exploration of SLiM-mediated interactions between globular domains of SARS-CoV-2 proteins and peptides found in the intrinsically disordered regions of the human proteome. We aimed to answer three main questions: (1) How widespread are the SLiM-based interactions of viral proteins?; (2) What are the human SLiM-containing binding partners of SARS-CoV-2 proteins?; and (3) Can we exploit the newly identified peptide binders as antiviral agents?.

Out of 26 SARS-CoV-2 protein domain constructs that we were able to express and purify, 11 showed phage enrichment from ProP-PD screens, and in total we identified 281 high/medium confidence human proteome-derived peptides that interact with SARS-CoV-2 protein domains. The results thus suggest that folded viral protein domains commonly bind to disordered regions of the human proteome. Whether the peptide-binding properties of the viral proteins have evolved

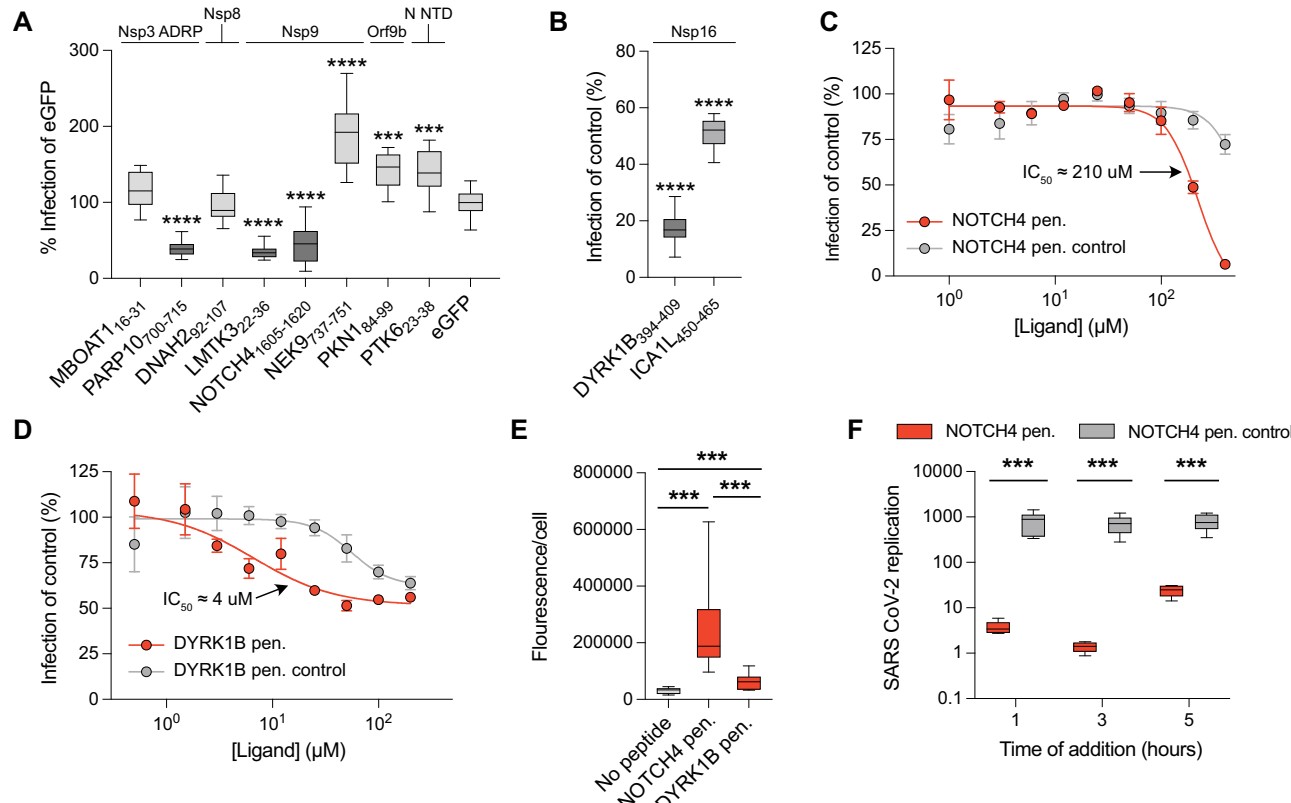

**Fig. 7 | Inhibition of SARS-CoV-2 viral infection propagation by lentiviral constructs or cell penetrating peptides. A** Inhibition is shown as percent infection of eGFP. Highlighted columns represent constructs, that showed significant inhibition. The target proteins of SARS-CoV-2 are stated above. Significance was determined using a two-sided unpaired $t$-test; *$p < 0.05$, **$p < 0.01$, ***$p < 0.001$, and ****$p < 0.0001$. Data are presented as box plots with mean, minimum, maximum, and interquartile range ($N = 12$). **B** Inhibitions by lentiviral constructs expressing Nsp16-targeting peptides are shown as percent infection of control. In the control construct the key interacting residues are mutated to alanine. A complete list of the lentiviral constructs can be found in Supplementary Data 7. Significance was determined same way as in (**A**). Data are presented as box plots with mean, minimum, maximum, and interquartile range ($N = 9$). **C** A serial dilution of either the cell penetrating NOTCH4 peptide (NOTCH4 pen.; red dots) or the cell penetrating NOTCH4 control peptide (NOTCH4 pen. control; gray dots) was added to SARS-CoV-2 infected VeroE6 cells. The $IC_{50}$ for the cell penetrating NOTCH4 peptide is indicated. Data is represented as mean ± SEM ($N \geq 3$; see Source Data for exact number of replicates for each peptide concentration). **D** The same experiment was repeated but with either the cell penetrating DYRK1B peptide (red dots) or the cell penetrating DYRK1B control peptide (gray dots). Data is represented as mean ± SEM ($N = 3$ for 0.5 µM, 1 µM, 100 µM and 200 µM and $N = 6$ for all other peptide concentrations). **E** Uptake of the Tat-tagged peptides. Total Tat fluorescence per cell. Data are cumulative of 12 stitched images (cell numbers: No peptide, $N = 821$, DYRK1B, $N = 891$, NOTCH4, $N = 624$) and is presented as box plots with mean, minimum, maximum, and interquartile range. Statistical significance was determined using GraphPad prism and a two-sided unpaired $t$ test; ***$p < 0.001$. **F** SARS CoV-2 replication in VeroE6 cells treated with 300 µM NOTCH4 pen. or NOTCH4 pen. control peptides. RNA replication is presented as fold induction compared to input. Data are cumulative from two independent experiments done in triplicates ($N = 6$) and are presented as box plots with mean, minimum, maximum, and interquartile range. Statistical significance was determined using GraphPad prism and a two-sided unpaired $t$ test; ***$p < 0.001$.

to serve a specific function during viral infection is uncertain. Most of the affinities measured for the host-virus PPIs were in the mid-micromolar range, which is within the typical range of SLiM-based interactions[71]. However, while these interactions may occur in the infected cell, it is plausible that the peptide-binding pockets of folded viral protein domains primarily evolved to bind proteins from the viral proteome (e.g. Nsp5, which cleaves Orf1a and b, and Nsp3 binding to N). The interactions with human proteins could be serendipitous and neutral for viral fitness, but such promiscuity might sometimes result in new "moonlighting" functions for evolution to act on. Notably, several of the interactions between SARS-CoV-2 proteins are mediated through SLiMs and facilitate correct localization (Nsp3-N)[11], complex formation (Nsp9-Nsp12)[24] or regulate the activity of other viral proteins (Nsp10-Nsp16)[72,73]. Our results show that the same viral protein domains, which facilitate interactions between viral proteins, can also engage in interactions with human SLiMs. This may result in competition between host and viral SLiMs for binding to Nsp3, Nsp9, and Nsp16, potentially impacting the rate of crucial infection processes such as viral RNA capping.

In case of Nsp1, the analysis identified more than 40 potential peptide ligands, although none of them reached a higher confidence level than 2 (of 4), and it was not possible to identify a common consensus motif among them. The results could be indicative of low affinity interactions to one or more sites on the protein. Of the five peptides selected for validations, two were among the most enriched ligands, and one was found in a previously reported interaction partner of Nsp1. Of the peptides tested, four did not bind within the concentration range tested, which was limited by the low solubility of the protein, and one appeared to bind with low affinity, supporting the observation that the selections enriched low affinity ligands.

With the caveats described above, our results expand the current knowledge on the SARS-CoV-2 host-virus interactome with detailed information on viral binding domains and defined binding sites in human proteins. In case of Nsp3 SUD-M and Nsp9, we also validated the interaction with the full length PRDM14 and NEK9 respectively, using pulldown experiments. We found for example that Nsp9 can bind to several human proteins that contain GΦxΦ[GDP] motifs, many

of which are associated with transcriptional regulation, suggesting a possible biological function of these interactions during the viral life cycle. It should however be noted that while the interactions we find can occur at the domain-peptide level and may occur in the context of the full-length proteins, the results do not provide any direct evidence for their relevance during viral infection.

While the discovery of potential novel viral binding SLiMs, such as the GΦxΦ[GDP] motif, in human proteins is intriguing, it is not clear which human globular domain(s), if any, bind to these motifs. Similarly, the C-terminal domain of the Ebola VP30 protein has previously been shown to bind to PPxPxY-containing peptides found in both viral (N) and human proteins[12,74]. As with the Nsp9 binding GΦxΦ[GDP] motif, it is not yet known which human protein(s) binds to the PPxPxY motif in a functional context. The discovery of such human SLiMs binding to viral proteins can provide an alternative starting point in the search for novel SLiM binding domains and thus contribute to our understanding of human SLiM-based interactions.

Since the start of the pandemic, there has been a rush to characterize SARS-CoV-2 protein interactions. Many of the large-scale studies have employed techniques such as affinity-purification coupled to mass spectrometry[5,39,42,64,75–77] that provide information on binary interactions as well as on larger complexes. Furthermore, several studies aimed to identify potential inhibitors of viral infection, but limited their scope to inhibition of RNA-dependent-RNA polymerase[78–81] and the proteases[82–84], or producing antibodies against spike protein[85–89]. To increase the likelihood of successful drug development, efforts need to be extended to include other SARS-CoV-2 proteins. The Nsp9 dimer interface with the GxxxG motif at its core has been proposed as a valid target for inhibitor development[90,91]. Our results established that the Nsp9 binding peptides from LMTK3 and NOTCH4 have antiviral effects, confirming the validity of targeting Nsp9. The coronavirus Nsp10/Nsp16 interaction has also been shown to be a valid target for antiviral peptides, using Nsp10-derived peptides[72,73,92]. Similarly, we found that our Nsp16 binding peptides had an antiviral effect. Other strategies targeting the Nsp10/Nsp16 complex include methyl donor site[68,90] and RNA binding site targeting, as reviewed recently[93].

In summary, we have demonstrated that ProP-PD screening is a viable strategy for identification of human peptides that bind to the globular domains of the SARS-CoV-2 proteome. It can be speculated that the interactions may influence the viral life cycle. We also showed that a subset of identified ligands inhibited RNA replication in cell culture, and that these peptides could be successfully converted into cell-penetrating anti-viral inhibitors. Thus, our study expands the available peptide repertoire that may be used as starting point for future drug discovery targeting coronaviruses.

## Methods

### Protein expression and purification

The cDNAs encoding SARS-CoV-2 protein domains were ordered from Genescript in pETM33 expression plasmids (Supplementary Data 1). The plasmid encoded an N-terminal His-tagged GST, a PreScisson protease cleavage site and the SARS-CoV-2 protein of interest. The proteins were expressed in BL21(DE3) gold *E. coli*. Bacteria was grown in 2YT medium (16 mg/ml peptone, 10 mg/ml yeast extract, 5 mg/ml NaCl) supplemented with 50 µg/ml kanamycin until OD$_{600}$ reached 0.6, when the protein expression was induced by the addition of 0.5 mM isopropyl β-D-1-thiogalactopyranoside (IPTG). Proteins were expressed overnight at 18 °C, bacterial cultures were harvested by centrifugation (4500 × *g*, 10 min) at 4 °C and resuspended in lysis buffer A (50 mM Tris/HCl pH 7.8, 300 mM NaCl, 10 µg/ml DNase I and RNase, 4 mM MgCl$_2$, 2 mM CaCl$_2$ and cOmplete EDTA-free Protease Inhibitor Cocktail). For ProP-PD selections, protein domains were purified on a GSH affinity column (Pierce glutathione agarose) according to the manufacturer's instructions. Following

elution with 10 mM GSH in buffer A, the sample was used in selections where the His-GST moiety was immobilized on the plate according to the protocol described previously[36]. For fluorescence polarization experiments, after the initial GSH affinity purification step, the His-GST was cleaved off by PreScission protease (1:100 dilution; produced in-house) overnight at 4 °C. Following the cleavage, the samples were applied on a nickel Sepharose excel resin column to remove the His-GST tag. The SARS-CoV-2 protein was recovered in unbound fraction from the nickel column. The purity of the samples was inspected by SDS-PAGE and if needed an additional purification step was introduced, where the SARS-CoV-2 protein sample was applied to a size-exclusion chromatography column (Superdex 75, Cytiva). The identity of pure protein samples was confirmed with matrix-assisted laser desorption/ionization time-of-flight mass spectrometry (MALDI-TOF MS). Finally, protein samples were dialyzed against 50 mM potassium phosphate buffer pH 7.4 and flash frozen until further use.

For NMR experiments the Nsp9 expression construct was expressed in M9 minimal medium containing 1 g/l $^{15}$NH$_4$Cl. After OD$_{600}$ reached 0.6 the protein expression was induced with 1 mM IPTG, and expressed overnight at 18 °C. After expression, the purification protocol was the same as described above.

### Proteomic peptide phage display

We recently published the design of a ProP-PD library expressing disordered regions found in the human proteome and a pipeline to analyze data from deep sequencing of enriched phages[36,94]. Briefly, GST-tagged bait proteins were immobilized in a 96-well Flat bottom Nunc MaxiSorp plates (Nunc, Roskilde, Denmark) at 4 °C for 18 h (10 µg per protein in 100 µl PBS). GST was immobilized in adjacent well as control. After immobilization the wells were blocked with blocking buffer (0.5% (w/v) BSA in PBS) for 1 h at 4 °C and washed 4 times with PT buffer (PBS supplemented with 0.05% Tween20). The phage library (prepared according to Benz et al.[36]) was added (100 µl in PBS supplemented with 1 mM DTT) to the wells with immobilized GST for 1 h, to remove nonspecifically binding phages, before being transferred to the wells with bait proteins. After 2 h the unbound phages were removed by washing the wells 5 times with 200 µl of PT buffer, and the bound phages were eluted by the addition of 100 µl of E. coli OmiMAX in log phase with subsequent incubation for 30 min at 37 °C. Bacteria were hyperinfected by the addition of M13KO7 helper phage and further incubated for 45 min at 37 °C. After successful infection, 100 µl of bacteria were transferred into 1 ml of 2×YT media supplemented with 100 µg/ml carbenicillin, 30 µg/ml kanamycin and 0.3 mM IPTG, followed by incubation for 18 h at 37 °C under agitation. Next, bacteria were pelleted at 2000 g for 10 min and phage supernatant was transferred to a fresh 96-well plate, where the pH was adjusted by the addition of 1/10 volume of 10× PBS and 1 mM DTT (final concentration), and heat inactivated at 65 °C for 10 min. The resulting phages were used next day as in-phages in the second round of selection, where the whole procedure was repeated. To ensure high enrichment of binding phages the selection procedure was repeated four times. After the final day of selection, the phage enrichment was evaluated by pooled phage ELISA in 384-well Flat bottom Nunc MaxiSorp plates (Nunc, Roskilde, Denmark). Bait proteins and GST control were immobilized (5 µg in 50 µl PBS per well) at 4 °C for 18 h followed by blocking of the remaining well surface with 100 µl of 0.5% BSA in PBS, at 4 °C for 1 h. Enriched phages from third and fourth rounds of selections (50 µl) were incubated with the corresponding bait protein for 1 h and the unbound phages were washed away four times with 100 µl PT buffer. Bound phages were incubated for 1 h with 50 µl of M13 HRP-conjugated M13 bacteriophage antibody (Sino Biological Inc; 11973-MM05T-H; 1:5000 diluted in PT, 0.05% BSA). The wells were again washed 4 times with PT buffer followed by one wash with PBS. Substrate was added (40 µl of TMB substrate, Seracare KPL) and the

enzymatic reaction was quenched by the addition of 40 μl of 0.6 M sulfuric acid. Finally, the absorbance at 450 nm was measured to quantify the phage enrichment with an iD5 plate reader (Molecular Devices).

## NGS analysis and ProP-PD data analysis

Peptide-coding regions were PCR-amplified and barcoded from the binding-enriched phage pools (5 μL) using Phusion High-Fidelity polymerase (Thermo Scientific). PCR products were normalized using Mag-bind Total Pure NGS, purified from a 2% agarose gel (QIA-quick Gel extraction Kit), and analyzed using Illumina MiSeq v3 (1 × 150 bp read setup, 20% PhiX). Reads were demultiplexed, adapter and barcode regions were trimmed, and sequences were translated into peptide sequences. Peptides were annotated using PepTools. Confidence levels were assigned based on four different criteria: occurrence in replicate selections, identification of overlapping peptide sequences, high counts, and occurrence of sequences matching consensus motifs, as previously outlined[36].

Position-specific scoring matrices were generated based on the peptide sets with confidence scores 2–4, using the SLiMFinder algorithm as implemented in PepTools[36,37].

Protein interaction networks were built with CytoScape 3.9.1[95]. Inkscape 1.2.1 (https://inkscape.org) and Python Matplotlib 3.5.1 library (https://matplotlib.org) were used to build the figures. For the comparison of ProP-PD results with other protein–protein interaction datasets the previously reported evidence of protein–protein interactions was obtained from Biogrid (release 4.4.212)[96], VirHostNet (release 3.0)[97], IMEx (release 1.4.11)[98], IntAct (release 4.2.18)[99] and MINT (August 2022)[100].

## Classical GO term enrichment analysis

GO term enrichment was performed using PepTools (http://slim.icr.ac.uk/peptools/input)[36]. Medium/high confidence peptides were used as input and were mapped to the corresponding proteins on which a hypergeometric analysis was performed to identify enriched functional annotations.

## Fluorescence polarization

Fluorescence polarization experiments were performed as previously described in detail[101]. Briefly, peptides were ordered either unlabeled or as FITC-labeled constructs from GeneCust. First, $K_D$ of the FITC labeled peptide for a certain interaction was determined by varying the protein concentration at a constant FITC-peptide concentration and fitting a hyperbolic function to the data. Next, a displacement experiment was performed to determine $K_D$ for the unlabeled peptide. Labeled peptide (10 nM) was pre-incubated with protein such that approximately 50–60% of the labeled peptide was bound to protein (Nsp3 Ubl1: 150 μM, Nsp3 ADRP: 75 μM, Nsp3 SUD-M: 100 μM, Nsp9: 17.4–25 μM, and Nsp16: 7 μM). Unlabeled peptide was then added at increasing concentration to compete out the binding of the labeled peptide. All experiments were performed as at least three technical replicates. Results were analyzed in GrafPad Prism (version 9.4.1). From this data set, the sigmoidal dose-response fitting function was used to obtain IC$_{50}$ values. These IC$_{50}$ values were further converted to $K_i$ (=$K_D$) of unlabeled peptide as described by Nikolovska-Coleska et al.[102]. Note that the error in the $K_D$ of the labeled peptide is not taken into account in the calculation of the error of $K_i$ for the displacer peptides. This means that the affinity measurement errors for displacement experiments are systematically slightly underestimated. However, since the experiments were done in parallel the comparison of the $K_D$ values of the displacer peptides for the same globular domain is not affected by this error. Because of this source of error between different interactions we attempt to discuss affinities in comparison to one another rather than in terms of absolute values.

## Enzymatic activity assay

To assess the enzymatic activity of Nsp5 and the inhibitory effect of identified peptide ligands a FRET based assay was performed as described previously[103]. In short, 1:1 dilution series of the inhibitor peptides were made (highest concentrations were 460 μM and 342 μM for DLG$_{577–592}$ and SHROOM2$_{403–417}$, respectively). To each reaction 20 μM FRET substrate, DABCYL-Lys-Thr-Ser-Ala-Val-Leu-Gln-Ser-Gly-Phe-Arg-Lys-Met-Glu-EDANS (Bachem Holding AG, Switzerland), and 150 nM Nsp5 were added to start the enzymatic reaction. Fluorescence emission was monitored every minute for 1 h at 37 °C and the velocity of the reaction was determined by the increase of the fluorescence signal over time. The initial reaction velocities were normalized against the highest rate in each data set, and a sigmoidal dose response (variable slope) equation was used to obtain the IC$_{50}$ values. Experiments were performed in 25 mM HEPES buffer, pH 7.4 and the data were analyzed in GraphPad Prism (version 9.4.1).

## GST-pulldown assay

The GST pulldown assay was performed as described previously[7]. HEK293T (kind gift from Johan Eriksson, Uppsala University, Sweden) cells were cultured on 100 mm culture plates and transfected using TurboFect (Thermo Fisher Scientific with plasmids expressing N-Flag and C-HA tagged PRDM14 (a gift from Danny Reinberg (Addgene plasmid # 84362)), N-Flag tagged NEK9 (a gift from Noboru Mizushima (Addgene plasmid # 168269)), C-HA tagged LMTK3 (a gift from Markku Varjosalo (Addgene plasmid # 187773), N-Flag tagged ICA1L (pcDNA3.1 backbone; Genescript, Netherlands) and N-Flag tagged ICA1L mut (pcDNA3.1 backbone; Genescript, Netherlands). After 48 h the cells were collected and lysed in GST lysis buffer containing 25 mM Hepes-KOH (pH 7.4), 12.5 mM MgCl$_2$, 100 mM KCl, 0.1 mM EDTA, 10% glycerol, 0.1% NP-40, supplemented with protease inhibitor and incubated at 4 °C for 30 min under mild agitation. The cell debris was pelleted by centrifugation at 21,000 g for 15 min and the supernatant was incubated with GST-tagged proteins for 1 h, under mild agitation. The beads were washed three times with the GST lysis buffer and the proteins were eluted with SDS-PAGE loading dye. Samples were separated by SDS-PAGE and analyzed by western blotting (nitrocellulose membrane (Amersham, Protran) for 2 h at 4 °C, 200 mA). The membrane was blocked with Intercept blocking buffer (LI-COR) for 1 h and subsequently incubated with primary mouse anti-Flag (Sigma, M2, F1804; 1:1000 dilution in Intercept blocking buffer), mouse anti-HA (Biolegend, 901501; 1:1000 dilution in Intercept blocking buffer) or goat anti-GST (Pharmacia, 27-4577; 1:1000 dilution in Intercept blocking buffer) antibodies, overnight at 4 °C. The membrane was washed three times with PBS-T (PBS with 0.1% Tween 20) and incubated with fluorescent secondary antibodies (IRDye®, LI-COR) against anti-mouse (IRDye® 800CW Goat-anti-Mouse Antibody; LI-COR, 926-32210; 1:10,000 dilution in Intercept blocking buffer) or anti-goat (IRDye® 680RD Donkey-anti-Goat Antibody; LI-COR, 926-68074; 1:10,000 dilution in Intercept blocking buffer) for 30 min at room temperature. Finally, the membrane was washed again three times with PBS-T and scanned using Odyssey scanner (LI-COR) with Image Studio Version 5.2.

## Circular dichroism spectroscopy

To determine if Nsp9Δα was folded, CD was monitored between 200 and 250 nm with a 1 nm bandwidth, scanning speed 50 nm/min and data pitch 1 nm. Experiments were performed on a JASCO J-1500 spectrometer at 25 °C in 50 mM potassium phosphate, 1 mM TCEP, pH 7.4 at 40 μM Nsp9Δα. To evaluate protein stability the sample was gradually heated (1 °C per minute) and the CD signal was monitored at 228 nm. The data were analyzed in GraphPad Prism (version 9.4.1). A sigmoidal denaturation suggested that the Nsp9Δα protein was folded in the experimental buffer at 25 °C. Data was collected on JASCO J-1500 CD spectrometer using Spectra manager 2 (Version 2.13).

## Nuclear magnetic resonance spectroscopy

Before NMR experiments the [15]N Nsp9 sample was dialyzed into 50 mM potassium phosphate buffer pH 6,7 supplemented with and supplemented with 1 mM TCEP and 10% $D_2O$. The final concentration of the sample was between 175 and 225 μM. All NMR experiments were performed on a 600 MHz Avance Neo HD NMR spectrometer (Bruker) equipped with a 5 mm TCI cryogenic probe. All [1]H-[15]N TROSY HSQC spectra were recorded at 25 °C making use of the BEST pulse sequence with 512 points in the direct and 256 points in the indirect dimension. Two or four scans per datapoint were taken with the relaxation delay of 200 μs. Similar experiments were performed upon the addition of the Nsp9 binding peptide ligands. Final concentrations of NKRF$_{8-23}$, LMTK3$_{22-36}$, and NOTCH4$_{1605-1620}$ were 416 μM, 230 μM and 323 μM, respectively. All Spectra were processed with TopSpin 3.2 and subsequently analyzed by MestReNova 14.1.0. The chemical peak shifts of Nsp9 residues were assigned based on comparison with previous assignments[61,65]. The perturbation of the chemical peak shifts upon addition of the peptides were calculated using equation 1:

$$\sqrt{\left(\Delta\delta_H{}^2 + \left(\frac{\Delta\delta_N}{R_{scale}}\right)^2\right)}$$

Where $\Delta\delta_H$ is chemical shift change in the hydrogen chemical shift dimension and $\Delta\delta_N$ is the chemical peak shift change in the nitrogen chemical shift dimension upon addition of peptide. $R_{scale}$ is a scaling factor set to 6.5 as described before[104].

To determine T1 and T2 relaxation times, TROSY HSQC-based experiments were employed[105,106] (PMID: 10729271 Zhu, PMID: 22689066). The overall rotational correlation time $\tau_c$ was estimated from the ratio of T1/T2 times (Supplementary Data 6). The same Nsp9 - NOTCH4$_{1605-1620}$ sample was used as for the previous experiments. For Nsp9 T1/T2 measurements fresh sample of Nsp9 was used at the concentration of 221 μM. The molecular weight of the species present in the sample was further estimated according to the equation 2:

$$M_w \approx \frac{3.8}{2} \times \tau_c$$

as described previously[107–109].

## ColabFold predictions

ColabFold[48] was used to predict the binding of peptides to globular protein domains from SARS-CoV-2. ColabFold is based on AlphaFold2[110] and AlphaFold2-multimer[111]. While the confidence scores varied for the predicted peptide conformations, all predictions for the structures of globular domains were in excellent agreement with solved crystal structure models (backbone alignment root mean square deviation between 0.4 and 1 Å).

## Alanine scanning SPOT arrays

20-mer, N-acetylated peptides on cellulose membranes were ordered from JPT (PepSpots). The membranes were activated with 5 ml methanol for 5 min at room temperature and washed with 10 ml TBST (50 mM Tris, 137 mM NaCl, 2.7 mM KCl, pH adjusted to 8.0 with HCl, 0.05% Tween-20) three times for 3 min at room temperature. The membranes were then incubated with 10 ml blocking buffer (5% skim milk powder (Merck Millipore, 115363) in TBST) for 2 h at room temperature while rotated. The blocked membranes were incubated with concentrated GST-HA-tagged proteins of interest in blocking buffer overnight, at 4 °C, while rotated. After three quick rinses with ice-cold TBST, the membranes were incubated with HRP-conjugated anti-GST antibody (Cytiva, RPN1236; 1:3000 dilution) in blocking buffer for 1 h at 4 °C, while rotated. Following three quick washes with ice-cold TBST, the chemiluminescent readout was carried out using

ECL reagent (Clarity Max Western ECL substrate, 1705062, Bio-Rad) and ChemiDoc Imaging system (Bio-Rad). The acquired raw tiff images were analyzed using Image Studio Lite Ver. 5.2., and all values were normalized to the wild-type results and the mean ± SD were calculated.

## SARS-CoV-2 infection assay

Vero E6 (*Cercopithecus aethiops*) cells (ATCC, CRL-1586) were cultured in Dulbecco's modified Eagle's medium (DMEM) (Sigma) supplemented with 10% fetal bovine serum (FBS) (HyClone) and 100 units/ml penicillin G with 100 μg/ml streptomycin solution (Gibco) at 37 °C, 5% $CO_2$, humidified chamber. MRC5, human lung fibroblast (ATCC CCL-171) cells were cultured in MEM medium (Gibco) supplemented with 10% FBS, and 100 units/ml penicillin G with 100 μg/ml streptomycin solution (Gibco) at 37 °C, 5% $CO_2$, humidified chamber.

SARS CoV-2 (SARS-CoV-2/01/human2020/SWE accession no/ GeneBank no MT093571.1, provided by the Public Health Agency of Sweden), was grown in VeroE6 cells, and used at passage number four. Human coronavirus 229E (HCoV-229E, ATCC CCL-171) was grown and titrated in MRC5 cells and used at passage one.

VeroE6 cells were transduced using the indicated lentiviruses as previously described[6]. After 72 h of transduction cells were infected with SARS CoV-2 for 16 h with MOI: 0.05 at 37 °C, 5% $CO_2$. For peptide treatments cells were first infected with SARS CoV-2 MOI: 0.05 at 37 °C, 5% $CO_2$, after 1 h of incubation the inoculum was replaced with medium containing the indicated concentration of peptide and cells were incubated at 37 °C, 5% $CO_2$ for 16 h. After the infections, cells were fixed in 4% formaldehyde, permeabilized in 0.5% triton X-100 in PBS. Viral infected cells were revealed by staining using primary monoclonal antibodies directed against SARS CoV-2 nucleocapsid (SARS CoV-2, Sino Biological Inc., 40143-R001; 1:500 dilution) or primary monoclonal mouse antibodies J2 directed against dsRNA (HCoV-229E, Scicons 10010500; 1:500 dilution), and secondary antibodies either donkey anti-mouse (Invitrogen; a31570; 1:500 dilution) or donkey anti-rabbit IgG Alexa Fluor 555 secondary antibody (Invitrogen; a31572; 1:500 dilution). Nuclei were counterstained with DAPI. Total cell number and the number of infected cells/well were determined using a TROPHOS Plate RUNNER HD® (Dioscure, Marseille, France). Number of infected cells were normalized to DAPI count and presented as percentage infection of mutated control peptide for the lentivirus transduced cells or as percentage of mock treated cells in the case of peptide treatment. Results were analyzed in GraphPad Prism (version 9.4.1) and the sigmoidal dose-response fitting function was used to obtain IC$_{50}$ values.

## Cell-penetrating peptide uptake assay

VeroE6 cells were treated with cell-penetrating peptides for 3 h before fixation using 4% formaldehyde. After fixation, cells were permeabilized in PBS containing 0.5% Triton X-100 and 20 mM glycine. Peptides were then stained using FITC Rabbit polyclonal to HIV1 tat antibodies (1:200, ab43016, abcam). Nuclei were stained using DAPI (1 μg/mL). Images were acquired using a Leica SP8 Laser Scanning Confocal Microscope with a 63x oil objective (Leica) and Leica Application Suit X software (LAS X, Leica). A total of 12 images for each condition was obtained and total fluorescent signal and cell number was quantified in ImageJ/Fiji (version 1.53t).

## Time dependent inhibition assay

VeroE6 cells were infected with SARS CoV-2 (MOI:1) for 1 h at 37°C and 5% $CO_2$. Then inoculum was removed and replaced with DMEM supplemented with either 2% FBS, 300 μM NOTCH pen or NOTCH pen. control at the indicated time post infection. After 9 h of infection cells were lyzed and total RNA was isolated from cells using nucleospin rna xs (Macherey Nagel) according to the manufacturer's instructions. Hundred nanogram RNA was used for cDNA synthesis

using High-capacity cDNA Reverse Transcription kit (Thermo Fisher). SARS-CoV-2 RNA was quantified using qPCRBIO probe mix Hi-ROX (PCR Biosystems) and the following primers and probes: GTCATGTGTGGCGGTTCACT, CAACACTATTAGCATAAGCAGTTGT and FAM-CAGGTGGAACCTCATCAGGAGATGC-BHQ. GAPDH was used as a reference gene, detected by RT² qPCR Primer Assay (NM_001195426, Qiagen) and the qPCRBIO SyGreen mix Hi-ROX (PCR Biosystems). SARS CoV-2 replication was quantified as SARS CoV-2 RNA fold induction over input RNA. All experiments were run on a StepOnePlus real-time PCR system (Applied Biosystems).

### Reporting summary

Further information on research design is available in the Nature Portfolio Reporting Summary linked to this article.

## Data availability

The details of the library designs including the proteins, peptides and statistics are available at http://slim.icr.ac.uk/phage_libraries/human/, and were described previously (PMID: 35044719). The PDB structures from PDBid: 6wxd and 7jyy were used in this study. Source data for this study are provided with this paper. Source data are provided with this paper.

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

## Acknowledgements

This work was supported by the grants from the Swedish Foundation for Strategic research (Y.I., P.J.: SB16-0039), the Swedish Research Council (Y.I.: 2020-03380; P.J.: 2020-04395; A.Ö.: 2018-05851), the Knut and Alice Wallenberg Foundation (Y.I., P.J., and A.K.Ö. via Science for Life Laboratory, KAW 2020.0241, V-2020-0699) and a Cancer Research UK Senior Cancer Research Fellowship (N.E.D.: C68484/A28159). We thank the medical faculty Umeå University strategic research resource and the Laboratory for Molecular Infection Medicine Sweden for generous support (A.K.Ö.), and the Biochemical Imaging Center at Umeå University and the National Microscopy Infrastructure, NMI (VR-RFI 2016-00968) for assistance in microscopy. Sequencing was performed by the SNP&SEQ Technology Platform in Stockholm. The facility is part of the National Genomic Infrastructure (NGI) Sweden and Science for Life Laboratory and is also supported by the Swedish Research Council and the Knut and Alice Wallenberg Foundation. We used the NMR Uppsala infrastructure, which is funded by the Department of Chemistry - BMC and the Disciplinary Domain of Medicine and Pharmacy, Uppsala University. We thank Dario Akaberi, Åke Lundkvist, and Johan Lennerstrand for providing active Nsp5 and giving advice on the enzymatic activity assay.

## Author contributions

F.M., C.B., N.E.D., P.J., and Y.I. conceived the study. F.M. performed FP experiments. C.B. performed phage selections. E.K. performed SPOT arrays, R.L. performed viral assays, RI produced lentiviruses, H.A., E.A., C.B., and F.M. produced proteins, F.M. and C.N.C. conducted NMR experiments and analyzed data. F.M., L.S., C.B,. N.E.D., A.K.Ö., P.J., and Y.I. analyzed data. F.M. and P.J. wrote the first draft. P.J. and Y.I. coordinated the study.

## Funding

## Competing interests

A patent application for protection of cell-penetrating peptides targeting Nsp9 and Nsp16 has been submitted under No. 2251153-9 and is currently under consideration (F.M., C.B., E.K., R.L., A.K.Ö., P.J., and Y.I.). The remaining authors declare no competing interests.
