## [Peer Review File · Nature Communications]

REVIEWER COMMENTS

Reviewer #1 (Remarks to the Author):

This study by Mihalic et al explores peptide-motif mediated protein-protein interactions (PPI) of SARS-CoV-2. In short, the authors deploy the ProP-PD methodology that their lab is known for and screen a range of SARS-CoV-2 proteins against a library of peptides derived from the human proteome. They find a range of peptides binding to viral proteins including Nsp3, Nsp9 and Nsp16 and confirm the binding for some of them using fluorescence polarization and map binding determinants using SPOT peptide arrays. Finally, they show that some of their peptides can disrupt viral complex formation as well as inhibit viral infectivity.

While there have been some work of peptide-motif mediated PPIs in viruses (including by the same lab), most previous work has focussed on peptide-motifs on the virus side; this is, to the best of my knowledge, the first systematic survey of peptide-motif binding capability of SARS-CoV-2 viral proteins. It is also a pretty comprehensive study, with a variety of data confirming their findings. I thus think this paper will be of interest to many people.

I just have a few comments for the authors:

1) I think it would be beneficial for at least a subset of the found interactions to confirm that the virus protein in question does actually interact with the human source protein of the peptide found in the phage display (e.g., in a pulldown); as far as I could tell, the authors only did a few select domains in FP. It would also make sense to try to model the full-length PPIs using AF2-multimer, which will perform better on full protein-protein interactions rather than just peptides.

2) I'm finding a bit odd that the DYRK1B-derived peptide inhibits infection quite efficiently as lentiviral construct, but seems to have much less efficacy as a tat peptide. One explanation may be that it has to do with uptake, which would be easy to check. On the same note, the authors may want to comment using their collabfold model whether it makes sense that FITC leads to an improvement in affinity. If so, there should be relatively obvious ways for optimization.

3) There are some peptides that seem to improve viral infectivity, which is interesting and could be explored or at least discussed a bit further.

As a minor comment, there are relatively few papers on linear motifs mediated interactions in SARS-CoV-2 and the authors only seem to cite their own.

Also, not to be too nitpicky, but some of the authors binding curves seem to be short of saturation, so it's probably worth noting that the Kds derived from these will have some larger uncertainty associated with them.

Why is the figure label on Fig 6A different from 6B?

Reviewer #2 (Remarks to the Author):

In this study the authors have used 26 SARS-CoV-2 proteins or their (known or predicted) subdomains of to screen their recently described (Benz et al, Mol Syst Biol 2022) phage library displaying peptides covering unstructured regions of the human proteome.

281 peptides considered "high" (8) or "medium confidence" (273) ligands were identified, most of which targeted Nsp9 (118), Nsp1 (47) or the protease domain of Nsp5 (32). Some of the identified peptide interactions (18/281) are supported by previously reported interactions, but in general the overall with other studies on SARS-CoV-2 host cell protein interactions is low.

Binding affinities that were measured for some of the peptides using fluorescence polarization were typically modest, with a few in the single digit micromolar range but mostly much weaker or too low to be measured. Consensus sequences could be established for the peptides binding to Nsp5 ([FLM][HQ][AS]; resembling the preferred cleavage site of its substrates) and to Nsp 9 (G[FL]xL[GDP]; resembling its known dimerization motif). Peptide scanning studies could also reveal key residues in peptides binding to Nsp10 or to the Ubl1, ADRP or SUD-M domains of Nsp3.

NMR studies indicated that the Nsp9-binding peptides could perturb packing of the Nsp9 hydrophobic core and thereby interfere with dimer formation. Eleven peptides targeting Nsp9, Nsp16, or ADRP of Nsp3 were expressed in VeroE6 from lentiviral constructs as four tandem repeats fused to GFP, and in five cases a modest inhibition of SARS-CoV-2 replication in these cells was observed. One Nsp9-binding and one Nsp16-binding peptide also showed some antiviral effect against SARS-CoV-2 when fused to an Arg-rich HIV-1 Tat sequence and tested as cell penetrating peptides.

This large study has been conducted in a professional manner and the extensive data reported are mostly of high quality. However, none of the results obtained are conceptually new or otherwise unusually exciting.

My major technical criticism relates to the studies on inhibition of SARS-CoV-2 replication by the identified peptides. More work and additional controls would be needed to convincingly demonstrate

that the observed effects are caused by specific binding of the peptides to their viral targets rather than off-target effects or toxicity. Lack of inhibition of a relevant control virus should be shown. Although some peptide controls seem to be included, they are not described sufficiently to understand their relevance. In addition to peptides mutated in the key consensus residues, rescue via overexpression of the viral target protein could be attempted. Also, mapping the inhibitory action of the peptide roughly to the expected phase in SARS-CoV-2 life cycle (entry, RNA replication, translation, virion production) should be possible.

Minor points:

The implications of using large proteins vs. protein domains as baits should be discussed more, and “predicted modular domains” better defined. The abstract refers to a screen carried out with eleven folded domains of SARS-CoV-2 proteins whereas the results say that 26 “protein constructs” were used.

The poor overlap with other SARS-CoV-2 host cell protein interaction screens should be better discussed with references to key studies in this area

Reviewer #3 (Remarks to the Author):

In the present work, the authors perform proteome-wide discovery of interactions between folded domains from non-structural and accessory SARS-CoV-2 proteins and human SLiMs using proteomic peptide phage display (ProPD). They attempt to cover 31 domains spanning most ORFs of the viral genome. Of these, 26 domains were successfully expressed, and 11 domains yielded enriched SLiMs. The ProPD experiments were validated using biochemical binding assays and SPOT arrays as well as structural analysis using NMR and AlphaFold. Selected peptides mediate inhibition of viral replication in cell assays, which provides a starting point for the development of SARS-CoV-2 inhibitors.

This work is of high significance for advancing the understanding of SARS-CoV-2 infection mechanisms, and for developing SARS-CoV-2-targeted therapies. The amount of work is impressive, comprehensively covering a wide array of SARS-CoV-2 domains. The interactions discovered through Pro-PD are for the most part novel and complementary to those discovered through other large-scale approaches such as mass spectrometry, while having the additional advantage of mapping the interaction region in the disordered partner.

The work is very important, but some points should be addressed. These concern mainly the comprehensive testing of major interaction classes identified and improving the level of support and soundness of some conclusions of the work, mainly those involving motif definition and structural mapping of the interactions. As a secondary point, the writing is at times unclear or incomplete, specifically for the description of some of the results sections. Most sections could use clear final paragraphs outlining the global conclusions that can be drawn from the experiments presented.

1- One of the strengths of the work lies in the comprehensive identification of interactions across SARS-CoV-2 protein domains. In this regard, it was surprising that of the three most enriched domains: Nsp9 (118 hits), Nsp1 (47 hits) and Nsp5-MPro (32 hits) only Nsp9 was thoroughly followed up. Nsp1 yielded 47 hits yet only three peptides were tested that were non-binders and no additional validations were carried out. Nsp5 validations were not performed. Nsp5 is the main protease (Mpro), and the enriched motif is claimed to match the protease cleavage site, suggesting the motifs could target the enzyme active site. This would be straightforward to test. Also, if the peptides do bind the active site, do they act as substrates or inhibitors of the active site? For the sake of comprehensiveness, some validation assays for Nsp1 and Nsp5 should be included and discussed.

2- ProPD results: The Logos presented in Fig. 2B look weird and do not seem to correspond to the sequences shown. There is no y-axis and the horizontal line at the bottom of the logo is not explained. Most importantly, the Logos don't match the list of sequences shown below for Nsp5 and Nsp9. In Nsp5 Q is enriched at the central position of most peptides but H is shown in the logo at the same position, SA are enriched at position 3 but only S is present in the logo, FL are enriched at position 1 but no enrichment/IC signal is shown in the logo. For Nsp9, G at position 1, FLIV at position 2, LI at position 4 have no correlate in the logo and some letters i.e., position 3 are floating. For Logos, the letter stack is continuous, so I don't know what the empty spaces and floating letters are. Finally, the method used to detect enrichment of motifs should be made explicit.

3- Fig. 2c: The method used for GO enrichment analysis is not explained. How is the subset shown in the panel chosen? Please explain. The meaning of the q value mentioned in text and the p value shown in Table S3 are not explained, please clarify.

4- Fig. 2c: Some interactions shown with the highest confidence score (4) don't seem to have been validated experimentally. Were they validated and they yielded negative results? If not tested, what was the rationale for this choice? Also, some of the higher affinity ($\sim 10\mu\text{M}$) interactions identified are from medium/low confidence (score 2) hits. It would be useful if the authors comment on this result.

5- The rationale for the choice of the 27 interactions that were validated was not explained. For example (see point 16), the Nsp9 motifs validated are not the same as those shown in Fig.2B. A comment in the text about the choice of peptides would be useful.

6- ProPD results: With the availability of Alphafold, it would be useful to have a sense of the structural accessibility of the motifs identified in the human proteins, at least for the dataset of proteins where binding of peptide hits was validated experimentally. Also, are the putative SLiMs conserved in alignments of these proteins? This could be shown as supplementary information and used to complement the analysis of motif patterns. Additional numerical scores for accessibility of the rest of the hits could also aid in the future selection of candidates for follow-up.

7- Nsp1 results. It is one of most successful selections with 47 ProPD hits, but only 3 peptides were tested and reported not to bind, how were they selected? Higher confidence? Previous interaction partner? Only EIF3A seems to be tested from the known interactors (Fig. 2C). Also, being one of the most successful selections, if a motif could not be derived from the peptide hits, this should be discussed in text and the implications explained.

Nsp3 Ubl1:

8- NCOA2 is the first motif identified for the Nsp3 Ubl1 domain (Kd 26 μ M). However, a larger construct containing the NCOA2 motif binds with 10-fold lower affinity (200 μ M), close to the limit of detection of the displacement assay. Are there any known auto-inhibitory interactions that would explain this behavior? Most important, based on this result, the individual motifs of NCOA1 & 3 should be tested to support the claim that NCOA1 & 3 do not interact with Nsp3-Ubl1, since the lower affinity of the larger constructs used in the current work could lead to a false negative result.

9- Motif pattern: The SPOT array of the NCOA2 motif reveals the main determinants of binding to be the L residue at p1 and the Y residue at p5 defining an "LxxxY" motif. The authors claim this explains the lack of binding of NCOA1/3, where p5 is a valine. However, the influence of the Y \rightarrow V replacement was not assessed since the SPOT array involves Ala substitutions, nor were the individual NCOA1/3 motifs tested. Experiments to support this conclusion include mutating Y \rightarrow V in the NCOA2 motif and testing binding for wt NCOA1/3 motifs and for motifs with a V \rightarrow Y substitution. Also, the p1 and p5 positions could accept other hydrophobic/aromatic replacements (only Ala is tested in the SPOT array). In fact, the high affinity LALLL motif from SARC-CoV-2 N has L at the p5 position. An alignment of the NCOA1/2/3 motifs should be shown and analyzed to assess these specificity determinants and the analysis of the motif pattern improved.

10- Mapping of the binding site: The NCOA2 and NYNRIN peptides bind to the same site as shown by the competitive displacement experiments. This site is proposed to match the binding site of the LALLLL motif based on structural predictions using AlphaFold. While the AF model has reasonable quality, it is still a model and requires additional experimental support to draw a sound conclusion. The authors test displacement with full length N which contains the LALLLL motif plus an additional Ubl1 binding site. The displacement experiment shows an increase in signal upon adding N, suggesting that a ternary complex is forming, and so the presence of the additional binding site in N complicates the analysis of the experiment. The claim that all peptides bind to the same cleft in Ubl1 need additional support from one of two experiments: 1) showing that the isolated LALLLL motif can compete for binding of the NCOA2 motif to Ubl1 and/or 2) performing a mutation on the proposed Ubl1 binding cleft. Also, the structure of the LALLLL motif bound to Ubl1 should be shown overlaid with the AF predictions for the peptides identified in this work.

Nsp3 ADRP:

11- Line 219-222: “The peptide SPOT array showed a strong signal for binding of the MBOAT116-31 peptide to Nsp3 ADRP, suggesting that the central 21-HPLSE-25 residues are critical for binding (Figure 3C, residues in bold).” The description of the results is unclear: the SPOT array shows that mutation of each residue within the HPLSE motif strongly decrease binding, which indicates that this region is the main determinant for binding. This should be stated more clearly in the results text.

12- Structural mapping of the binding site: If the predictions are low confidence (score ~30), they can't be used to make claims about the binding site in the absence of supporting mutagenesis of the ADRP binding cleft and/or direct competition experiments with motifs known to bind to the proposed site. If no additional experiments are performed, showing the models in main text figures should be avoided and the conclusions of the paragraph below amended to indicate that while both peptides seem to bind to the same region in ADRP, no clear conclusions can be drawn regarding the binding site:

The ColabFold predictions of the Nsp3 ADRP binding peptides did not converge with high confidence (pLDDT of ~30; Figure S3). However, manual inspection indicated that the peptide binding was restricted to the surface between the N-terminal beta sheet (β 1) and the C-terminal alpha helix (α 6) (Figure 3B, Figure S3)”

Nsp3 SUD-M:

13- Same as point 12 for ADRP, the proposed binding site obtained with the AF prediction is of moderate quality and should be validated with mutagenesis of the SUD-M binding cleft. If no additional mapping experiments are performed, showing the AF models in main text figures should be avoided and the conclusions of the paragraph amended.

14- Lines 241-245: The agreement between the relevant residues identified in the SPOT array and the face of the helix facing the domain, should be better explained in text and the residues labeled in the structure of SUD-M shown in Fig. 3B to facilitate the analysis.

15- For the conclusions of the Nsp3 section, depending on the additional experiments performed, the writing should be improved to discriminate between what was learnt of the mapping the binding sites in the human proteins and determining the motif specificity, which is different from mapping (or not) of the binding site on the different domains, with the latter requiring additional support.

Nsp9

16- The set of peptides shown & tested in Fig. 4B are different from the set of peptides shown in Fig.2B. Please explain the selection criteria. Also, the authors should explain how the regular expression for the enriched motif was derived.

17- The authors adjust the motif pattern to GPhixPhi[GDP] based on the SPOT array performed with NKRF. However, this pattern is not supported by the SPOT array of LMTK3: Phe→Ala substitution at p2 yields no change in binding and Gly→Ala substitution at p5 strongly increases binding. The LMTK3 results need to be discussed and the conclusions amended. Unless additional mutagenesis experiments are performed, the pattern can only be claimed for NKRF, and a different binding specificity is observed in LMTK3 suggesting a broader binding pattern. Mutations following the identified motif increase binding for both peptides, and this result also needs to be discussed and interpreted.

18- Nsp9 can exist as a monomer (i.e., when forming part of the Nsp7/8/9/12 RNA capping complex) and can form a homodimer stabilized by a dynamic C-terminal helix. The authors propose that since the peptide pattern resembles the Nsp9 C-terminal helix involved in dimerization, these peptides could bind to the dimerization interface and interfere with dimer formation. Since the AF predictions do not converge on a common binding site for different peptides, the authors perform NMR experiments to map the binding site on Nsp9 using three peptides. The first analysis is the chemical shift perturbation, which is mapped on the Nsp9 structure (Fig. 4C/E). The mapping of the binding site to the dimer interface is complicated by the fact that the C-terminal helix peaks are missing (as previously reported)

and multiple other sites across Nsp9 loops and the hydrophobic core show changes in CS. The authors claim very similar CS perturbation patterns for the three peptides that support a common binding mode, but this would be more clearly evidenced by showing per-residue I/I₀ plots and CS plots for the three complexes. Please add this information to the supplementary info and possibly Main Figure: are other surface residues identified? This first set of experiments does not achieve unequivocal mapping of the binding interface.

19- Another piece of evidence is from analyzing whether addition of the peptides induces dissociation of the Nsp9 dimer. However, this is done by MW estimation from the T₁/T₂ relaxation times, which have a contribution from the molecule MW and shape and is therefore not unequivocal. Free Nsp9 has $t_c=15.3\text{ns}$ ($MW_{app} = 29\text{ kDa}$) and motif bound Nsp9 has $t_c=9.9\text{ns}$ ($MW_{app} = 18.8\text{ kDa}$). While the results suggest free Nsp9 is dimeric and bound Nsp9 is monomeric, there are other explanations (e.g., free Nsp9 could be an extended monomer). A direct proof for the oligomeric state can be obtained by performing SEC-MALS or SEC-SLS measurements of free and NOTCH4-bound Nsp9, allowing an unequivocal determination of the MW of the species.

20- To further test whether the peptides bound through the C-terminal helix the authors used a delta-Ct Nsp9 construct. This construct involves a 20-aa deletion that may disrupt Nsp9 folding and/or cause non-native oligomerization that confounds the result of the binding experiment. Therefore, biophysical characterization (Far-UV CD and SEC or SEC-SLS/MALS) should be performed to verify that the construct retains a native fold and secondary structure, and that it is a monomer. Without these controls, the experiment cannot be used to claim the peptides bind to the C-terminal helix region.

21- Alternatively, several point mutations that stabilize the Nsp9 monomer state have been reported, and they could be used for mapping the proposed interaction site.

Conclusions section:

22- Based on additional evidence obtained, the discussion should clearly review what information was gained for each domain-SLiM interaction. The claim that the novel human SLiMs bind to known binding clefts in the SARS-CoV-2 domains depend on additional experiments and should otherwise be amended.

23- It would be useful to include a broader discussion on the possible role of the human SLiMs and the domains they target in human cells. Did the authors perform structural mapping studies to identify possible folds in human proteins that resemble the SARS-CoV-2 domains and could be targeted by these SLiMs? Are the human SLiMs conserved in the identified proteins?

Minor points:

- Fig. 2: all panels must be cited specifically in the main text (only figure and not figure panels are cited).

- Fig. 3a legend says the data are presented as means \pm SD but the n is not defined. Kd values vary slightly with respect to Table S4, correct. Figure 3c, SPOT arrays define n value.

- Figure 3B: Label the residues shown as sticks so they can be identified in the sequences of the motifs shown in Fig. 3c.

- line 197/198: "This finding clarified the lack of binding of NCOA1 and NCOA2 paralogues" should say "This finding clarified the lack of binding of NCOA1 and NCOA3 paralogues". However, this claim should be further supported by binding of individual motifs.

- Line 475 Discussion "mass spectroscopy" is probably "mass spectrometry"

- Line 480-481: "The Nsp9 dimer interface in the C-terminal helix with the GxxxG motif at its core has been proposed as a valid target for inhibitor development". The sentence is unclear, rephrase.

- Fig. S3 the line in the pIDDT plots probably represents the break between the domain and peptide in the prediction but this is not explained. Explain in figure legend.

- Fig S4: if these are duplicate experiments explain. Provide raw SPOT array data.

Reviewer #1 (Remarks to the Author):

This study by Mihalic et al explores peptide-motif mediated protein-protein interactions (PPI) of SARS-CoV-2. In short, the authors deploy the ProP-PD methodology that their lab is known for and screen a range of SARS-CoV-2 proteins against a library of peptides derived from the human proteome. They find a range of peptides binding to viral proteins including Nsp3, Nsp9 and Nsp16 and confirm the binding for some of them using fluorescence polarization and map binding determinants using SPOT peptide arrays. Finally, they show that some of their peptides can disrupt viral complex formation as well as inhibit viral infectivity.

While there have been some work of peptide-motif mediated PPIs in viruses (including by the same lab), most previous work has focussed on peptide-motifs on the virus side; this is, to the best of my knowledge, the first systematic survey of peptide-motif binding capability of SARS-CoV-2 viral proteins. It is also a pretty comprehensive study, with a variety of data confirming their findings. I thus think this paper will be of interest to many people.

Reply

We thank the reviewer for the appreciating our study and providing helpful feedback.

Comment

1) I think it would be beneficial for at least a subset of the found interactions to confirm that the virus protein in question does actually interact with the human source protein of the peptide found in the phage display (e.g., in a pulldown); as far as I could tell, the authors only did a few select domains in FP. It would also make sense to try to model the full-length PPIs using AF2-multimer, which will perform better on full protein-protein interactions rather than just peptides.

Reply

We agree with the reviewer and have attempted the pulldown experiments of PRDM14 with Nsp3 SUD-M, LMTK3 and NEK9 with Nsp9 and ICAIL/ICA1L mut with Nsp16. Of the five cases tested, two pulldowns were successful, namely NEK9 with Nsp9 and PRDM14 with Nsp3 SUD-M (see figure below). See **Figure 3C** and **Figure 5G** as well as **Figure S12** for further details. We have now included pulldown results in the revised manuscript. We also attempted ColabFold modeling of full-length human proteins with SARS-CoV-2 domains, but the resulting predictions did not converge on the corresponding human peptide:SARS-CoV-2 protein domain interaction. While a positive result (i.e., the region

corresponding to the ProP-PD-selected peptide interacts with the domain) would corroborate our conclusions, a negative result is inconclusive, and therefore we chose to not show these models.

Comment:

2) I'm finding a bit odd that the DYRK1B-derived peptide inhibits infection quite efficiently as lentiviral construct, but seems to have much less efficacy as a tat peptide. One explanation may be that it has to do with uptake, which would be easy to check. On the same note, the authors may want to comment using their collabfold model whether it makes sense that FITC leads to an improvement in affinity. If so, there should be relatively obvious ways for optimization.

Reply:

Following the suggestion, we examined the uptake of the peptides and found indeed that the uptake of the DYRK1B-derived peptide is only a fraction of the uptake of the NOCTH4-derived peptide (see figure below). We have added the information to the manuscript and to the **Figure 7E**.

We thank the reviewer for the suggestion on how to optimize the peptide, which we may consider for future studies. For now, we note that the FITC-labeled peptides have 10-fold higher affinity for Nsp16 and DYRK1B. We inspected the model, and there is no clear hydrophobic patch on the surface of Nsp16 where the N-terminus of the peptide binds. Thus, we do not have a clear explanation as to why FITC increases the affinity for the interaction. Further optimization and development of tight ligands is beyond the scope of this study.

Comment:

3) There are some peptides that seem to improve viral infectivity, which is interesting and could be explored or at least discussed a bit further.

Reply:

We have added a comment on this in the results section. Two out of three peptides with pro-viral effects are weak binders of N-NTD and Orf9b, suggesting that the observed effects might be due to off-target effects that could potentially be linked to inhibition of the antiviral response.

Comment:

As a minor comment, there are relatively few papers on linear motifs mediated interactions in SARS-CoV-2 and the authors only seem to cite their own.

Reply:

We agree and have added additional references to the text to better cover the topic.

Comment:

Also, not to be too nitpicky, but some of the authors binding curves seem to be short of saturation, so it's probably worth noting that the Kds derived from these will have some larger uncertainty associated with them.

Reply:

The graphs showing saturation experiments are presented on a logarithmic scale, which might give a false impression of poor saturation. Below we show an example of a typical saturation binding experiment plotted on a logarithmic and linear scale, respectively, to illustrate the point. Moreover, since the affinities are mainly compared to one another for a certain interaction (in terms of fold change), low saturation will not change the conclusions in the manuscript.

Nsp3 Ub1: FITC-NCOA2 1074-1089 saturation

Nsp3 Ub1: FITC-NCOA2 1074-1089 saturation

Comment:

Why is the figure label on Fig 6A different from 6B?

Reply:

The labels on the y axis differ because different controls were used. In 6A the inhibition was compared to the transfection with eGFP expressing plasmid alone while in 6B it was compared to the transfection with eGFP construct that was fused to the peptide mutant version of the inhibitory peptides. For details see **Table S7**. Note that **Figure 6** in the initial manuscript is now **Figure 7** in the revised version.

Reviewer #2 (Remarks to the Author):

In this study the authors have used 26 SARS-CoV-2 proteins or their (known or predicted) subdomains of to screen their recently described (Benz et al, Mol Syst Biol 2022) phage library displaying peptides covering unstructured regions of the human proteome.

281 peptides considered “high” (8) or “medium confidence” (273) ligands were identified, most of which targeted Nsp9 (118), Nsp1 (47) or the protease domain of Nsp5 (32). Some of the identified peptide interactions (18/281) are supported by previously reported interactions, but in general the overall with other studies on SARS-CoV-2 host cell protein interactions is low.

Binding affinities that were measured for some of the peptides using fluorescence polarization were typically modest, with a few in the single digit micromolar range but mostly much weaker or too low to be measured. Consensus sequences could be established for the peptides binding to Nsp5 ([FLM][HQ][AS]; resembling the preferred cleavage site of its substrates) and to Nsp 9 (G[FL]xL[GDP];

resembling its known dimerization motif). Peptide scanning studies could also reveal key residues in peptides binding to Nsp10 or to the Ubl1, ADRP or SUD-M domains of Nsp3.

NMR studies indicated that the Nsp9-binding peptides could perturb packing of the Nsp9 hydrophobic core and thereby interfere with dimer formation. Eleven peptides targeting Nsp9, Nsp16, or ADRP of Nsp3 were expressed in VeroE6 from lentiviral constructs as four tandem repeats fused to GFP, and in five cases a modest inhibition of SARS-CoV-2 replication in these cells was observed. One Nsp9-binding and one Nsp16-binding peptide also showed some antiviral effect against SARS-CoV-2 when fused to an Arg-rich HIV-1 Tat sequence and tested as cell penetrating peptides.

This large study has been conducted in a professional manner and the extensive data reported are mostly of high quality. However, none of the results obtained are conceptually new or otherwise unusually exciting.

Comment

My major technical criticism relates to the studies on inhibition of SARS-CoV-2 replication by the identified peptides. More work and additional controls would be needed to convincingly demonstrate that the observed effects are caused by specific binding of the peptides to their viral targets rather than off-target effects or toxicity. Lack of inhibition of a relevant control virus should be shown.

Reply

We already included in the original submission a control experiment using the cell-permeable peptides and their influence on human coronavirus HCoV-229 infection, (current **Figure S15**). We did not observe any non-specific reduction in SARS-CoV-2 infectiousness.

Comment

Although some peptide controls seem to be included, they are not described sufficiently to understand their relevance.

Reply

We performed the key control experiment, showing that non-binding peptides with mutated motifs have no antiviral effect (**Figure 7C-D**). We have attempted a better explanation of the experiment in the text and in the figure legend.

Comment

In addition to peptides mutated in the key consensus residues, rescue via overexpression of the viral target protein could be attempted. Also, mapping the inhibitory action of the peptide roughly to the expected phase in SARS-CoV-2 life cycle (entry, RNA replication, translation, virion production) should be possible

Reply

We agree. Given the low cellular uptake of the DYRK1B derived peptide (see reply to reviewer 1) we focused on the Nsp9-binding NOTCH4 derived peptide and mapped the inhibitory action of the peptide to viral replication (see below), as expected based on the role of Nsp9 in the SARS-CoV-2 life cycle. The information has now been added to **Figure 7** and to the text.

*SARS CoV-2 replication in VeroE6 cells treated with either 300 μ M NOTCH pen. or NOTCH pen. control. Viral replication is presented as fold induction compared to input. Data are cumulative from 2 independent experiments done in triplicates ($N = 6$). Statistical significance was determined using GraphPad prism and an unpaired t -test; *** $p < 0.001$.*

Comment

Minor points: The implications of using large proteins vs. protein domains as baits should be discussed more, and “predicted modular domains” better defined. The abstract refers to a screen carried out with eleven folded domains of SARS-CoV-2 proteins whereas the results say that 26 “protein constructs” were used.

Reply

We have amended the text of the introduction such that the distinction between globular domains and full-length proteins is more clear:

In the present study we systematically investigated the virus-human protein-protein interactome of folded protein domains encoded by the SARS-CoV-2 genome and peptides representing the IDRs of the human proteome. Several of the SARS-CoV-2 proteins contain multiple domains, for example, Nsp3 consists of ten folded domains (Yan et al. (2022) *Signal Transduct Target*

Ther 7, 26). Here we focused on the folded domains rather than full length viral proteins to enable purification of bait proteins for phage display selections.

Furthermore, in the discussion we have added this text:

With the caveats described above, our results expand the current knowledge on the SARS-CoV-2 host-virus interactome with detailed information on viral binding domains and defined binding sites in human proteins. In case of Nsp3 SUD-M and Nsp9, we also validated the interaction with the full length PRDM14 and NEK9 respectively, using pulldown experiments. We found for example that Nsp9 can bind to several human proteins that contain GΦxΦ[GDP] motifs, many of which are associated with transcriptional regulation, suggesting a possible biological function of these interactions during the viral life cycle. It should however be noted that while the interactions we find can occur at the domain-peptide level and may occur in the context of the full-length proteins, the results do not provide any direct evidence for their relevance during viral infection.

Comment

The poor overlap with other SARS-CoV-2 host cell protein interaction screens should be better discussed with references to key studies in this area

Reply

We agree and have expanded the section and included relevant references (see reply to reviewer 1).

Reviewer #3 (Remarks to the Author):

In the present work, the authors perform proteome-wide discovery of interactions between folded domains from non-structural and accessory SARS-CoV-2 proteins and human SLiMs using proteomic peptide phage display (ProPD). They attempt to cover 31 domains spanning most ORFs of the viral genome. Of these, 26 domains were successfully expressed, and 11 domains yielded enriched SLiMs. The ProPD experiments were validated using biochemical binding assays and SPOT arrays as well as

structural analysis using NMR and AlphaFold. Selected peptides mediate inhibition of viral replication in cell assays, which provides a starting point for the development of SARS-CoV-2 inhibitors.

This work is of high significance for advancing the understanding of SARS-CoV-2 infection mechanisms, and for developing SARS-CoV-2-targeted therapies. The amount of work is impressive, comprehensively covering a wide array of SARS-CoV-2 domains. The interactions discovered through Pro-PD are for the most part novel and complementary to those discovered through other large-scale approaches such as mass spectrometry, while having the additional advantage of mapping the interaction region in the disordered partner.

The work is very important, but some points should be addressed. These concern mainly the comprehensive testing of major interaction classes identified and improving the level of support and soundness of some conclusions of the work, mainly those involving motif definition and structural mapping of the interactions. As a secondary point, the writing is at times unclear or incomplete, specifically for the description of some of the results sections. Most sections could use clear final paragraphs outlining the global conclusions that can be drawn from the experiments presented.

Comment

1- One of the strengths of the work lies in the comprehensive identification of interactions across SARS-CoV-2 protein domains. In this regard, it was surprising that of the three most enriched domains: Nsp9 (118 hits), Nsp1 (47 hits) and Nsp5-MPro (32 hits) only Nsp9 was thoroughly followed up. Nsp1 yielded 47 hits yet only three peptides were tested that were non-binders and no additional validations were carried out. Nsp5 validations were not performed. Nsp5 is the main protease (Mpro), and the enriched motif is claimed to match the protease cleavage site, suggesting the motifs could target the enzyme active site. This would be straightforward to test. Also, if the peptides do bind the active site, do they act as substrates or inhibitors of the active site? For the sake of comprehensiveness, some validation assays for Nsp1 and Nsp5 should be included and discussed.

Reply

We agree with the reviewer and have completed the set by testing additional Nsp1 and Nsp5-binding peptides, respectively. For Nsp1, we confirmed a low affinity interaction, and for Nsp5 we confirmed peptide binding to the catalytically inactive protein, as well as inhibition of catalytic activity when using

an active enzyme. We have added the information to the manuscript in a new **Figure 4**. For details also see **Figure S1** and **Table S4**.

Comment

2- ProPD results: The Logos presented in Fig. 2B look weird and do not seem to correspond to the sequences shown. There is no y-axis and the horizontal line at the bottom of the logo is not explained. Most importantly, the Logos don't match the list of sequences shown below for Nsp5 and Nsp9. In Nsp5 Q is enriched at the central position of most peptides but H is shown in the logo at the same position, SA are enriched at position 3 but only S is present in the logo, FL are enriched at position 1 but no enrichment/IC signal is shown in the logo. For Nsp9, G at position 1, FLIV at position 2, LI at position 4 have no correlate in the logo and some letters i.e., position 3 are floating. For Logos, the letter stack is continuous, so I don't know what the empty spaces and floating letters are. Finally, the method used to detect enrichment of motifs should be made explicit.

Reply

There seems to have been some information lost from the logos, which now should be corrected (e.g., the missing A in the logo, the floating letters etc). We have added an explanation that the method used was SLiMFinder as implemented in PepTools, and provided references. The dotted line indicated the p-value of 0.001. However, to avoid confusion we have removed the dotted lines from the revised figure. The peptides shown in the alignment do not represent all peptides on which the logos are built, but only a select subset. In particular, for Nsp5, we included peptides with an LQA motif matching the known proteolytic cleavage site of the enzyme. Full peptide sets are provided in supplemental Table 2.

Comment

3- Fig. 2c: The method used for GO enrichment analysis is not explained. How is the subset shown in the panel chosen? Please explain. The meaning of the q value mentioned in text and the p value shown in Table S3 are not explained, please clarify.

Reply

We agree and have now added a description of the GO term enrichment analysis to the method section, and have explained it better in the main text. The q was the *P*-values corrected for multiple hypothesis testing using Benjamini–Hochberg correction.

The interactors shown were selected to sample ligands with different confidence levels, and to include previously reported interactors. For Nsp9 we further included ligands that are associated with GO terms related to transcriptional regulation, as that was the main GO term enriched for the ligand set.

Comment

4- Fig. 2c: Some interactions shown with the highest confidence score (4) don't seem to have been validated experimentally. Were they validated and they yielded negative results? If not tested, what was the rationale for this choice? Also, some of the higher affinity (~10 μ M) interactions identified are from medium/low confidence (score 2) hits. It would be useful if the authors comment on this result.

Reply

This is correct. We did not exclude any data, but did not characterize all interactions with highest confidence score. Generally, the peptides were selected to include highly enriched ligands (based on NGS counts) that were specifically enriched for the baits. For Nsp9 there was a large number of peptides enriched, and we sampled ligands with a range of NGS counts. In addition, we included a couple of peptides with lower scores found in proteins previously reported to interact with the bait proteins (e.g., the PARP10 peptide binding to Nsp3 ADRP).

In part, the selections of peptides for further investigations were based on preliminary data, which explains for example why the two highest confidence ligands of Nsp9 were not tested. However, for ligands from the highest confidence level (4) we know by experience that they will bind the baits with close to 100% certainty, while the peptide sets from lower confidence levels may contain false positives. Thus, while our success rate would likely increase in terms of validations if we only focused on the highest confidence level, the characterization of the medium confidence ligands becomes in a way more valuable as it provides support for the notion that also this set contains true positive ligands.

Finally, lower confidence level peptides (or peptides with low NGS counts) could also be high affinity ligands, since the phage selections as well as the NGS analysis may be skewed by factors other than affinity. Such factors include phage biogenesis and stability, "stickyness" of tryptophane-containing peptides during phage selection (PMID: 21300698), as well as biases introduced during PCR amplifications of amplicons for NGS analysis. In conclusion, while we cannot follow up on all peptides, we argue that sampling from different confidence levels may provide valuable insights.

Comment

5- The rationale for the choice of the 27 interactions that were validated was not explained. For example (see point 16), the Nsp9 motifs validated are not the same as those shown in Fig.2B. A comment in the text about the choice of peptides would be useful.

Reply

We have added an explanation of the selection of peptides to the result section and discussed the issue under the previous comment. We have also updated figure 2B to include the validated peptides.

Comment

6- ProPD results: With the availability of AlphaFold, it would be useful to have a sense of the structural accessibility of the motifs identified in the human proteins, at least for the dataset of proteins where binding of peptide hits was validated experimentally. Also, are the putative SLiMs conserved in alignments of these proteins? This could be shown as supplementary information and used to complement the analysis of motif patterns. Additional numerical scores for accessibility of the rest of the hits could also aid in the future selection of candidates for follow-up.

Reply

The HD2 library design used IUpred predictions to identify intrinsically disordered regions to be included in the library design as AlphaFold2 was not available at the time (see Benz et al). We agree that AlphaFold2 can, to some extent, be used to prioritize ligands for follow ups, as it can be used to score surface accessible regions. Following the advice of the reviewer we performed such analysis on the validated cases (See new **Figure S14** where the region corresponding to the selected peptide is highlighted in the respective predicted structure). The predictions suggest that the peptide regions that were identified as viral ligands usually are present in disordered and solvent accessible parts of the protein, consistent with the IUpred predictions.

Regarding conservation, we would expect any motifs to be evolutionary conserved if they have a function in the human cell. Any conservation due to interactions with viral protein domains are unlikely unless the proteins are directly involved in antiviral defense. Even then conservation may not be expected since co-evolution between virus and host could take different directions as hosts diverge over millions of years and the viruses with them. RNA viruses such as SARS-COV-2 can obviously be very generalistic and jump between hosts. In any case, the question is interesting from the point of view of motifs in the human proteome. We have provided a new **Fig. S6**, which shows alignments for the regions containing the

peptide, from interactions that were validated. In some cases, the peptide sequences and putative motif are conserved among vertebrates, in other cases only in mammals or not at all.

Comment

7- Nsp1 results. It is one of most successful selections with 47 ProPD hits, but only 3 peptides were tested and reported not to bind, how were they selected? à higher confidence? Previous interaction partner? Only EIF3A seems to be tested from the known interactors (Fig. 2C). Also, being one of the most successful selections, if a motif could not be derived from the peptide hits, this should be discussed in text and the implications explained.

Reply

Nsp1 enriched for a large number of peptides, but among them none were highly enriched, and none reached a higher confidence level than 2. Furthermore, we could not identify a common consensus motif despite the large number of peptides. Based on this, we would judge the results as fairly weak, which could be indicative of low affinity interactions to one or more sites on the protein. Nevertheless, we selected 3 peptides for validations, two of which were among the more enriched ligands (the VIL1₂₁₅₋₂₃₀ peptide), and one that was found in a previously reported interaction partner of Nsp1 (the EIF3A₅₂₉₋₅₄₄ peptide). However, these peptides did not bind within the concentration range tested, which was limited to ~30 µM by the low solubility of the Nsp1 protein in our hands. During the revision we selected additional peptides for validation among the higher count ligands and could confirm weak binding of one of the ligands tested (KCNQ5₁₀₄₋₁₁₉), supporting that the selections against Nsp1 enriched for low affinity ligands.

We have added this information to the results and the discussion.

Comment

8- NCOA2 is the first motif identified for the Nsp3 Ubl1 domain (Kd 26µM). However, a larger construct containing the NCOA2 motif binds with 10-fold lower affinity (200 µM), close to the limit of detection of the displacement assay. Are there any known auto-inhibitory interactions that would explain this behavior? Most important, based on this result, the individual motifs of NCOA1 & 3 should be tested to support the claim that NCOA1 & 3 do not interact with Nsp3-Ubl1, since the lower affinity of the larger constructs used in the current work could lead to a false negative result.

Reply

We are not aware of auto-inhibitory interactions that could explain the behavior of the larger NCOA2 construct. We would rather lean towards an explanation where the regions flanking the peptide provide repulsion. In the native interaction between the paralog NCOA3 and CBP/p300, flanking regions of the longer construct have been shown to increase affinity 3-fold (PMID: 35605677). In fact, it is likely that most motif-mediated interactions are influenced by context to different extents.

Nevertheless, we agree with the reviewer's experimental suggestions and have tested the binding of the individual motifs of NCOA1 & 3. These peptides bound Nsp3 Ubl1 better than the longer construct, but still with low affinity. In addition, we tested if the mutation of the critical Y in the NCOA2 peptide to V would decrease binding, and if mutation of the V to Y in the NCOA1 & 3 peptides would increase and found this to be the case. We have added these new data to the manuscript, in the text and to the new **Figure S3**.

Comment

9- Motif pattern: The SPOT array of the NCOA2 motif reveals the main determinants of binding to be the L residue at p1 and the Y residue at p5 defining an “LxxxY” motif. The authors claim this explains the lack of binding of NCOA1/3, where p5 is a valine. However, the influence of the YàV replacement was not assessed since the SPOT array involves Ala substitutions, nor were the individual NCOA1/3 motifs tested. Experiments to support this conclusion include mutating YàV in the NCOA2 motif and testing binding for wt NCOA1/3 motifs and for motifs with a VàY substitution. Also, the p1 and p5 positions could accept other hydrophobic/aromatic replacements (only Ala is tested in the SPOT array). In fact, the high affinity LALLLL motif from SARS-CoV-2 N has L at the p5 position. An alignment of the NCOA1/2/3 motifs should be shown and analyzed to assess these specificity determinants and the analysis of the motif pattern improved.

Reply

We agree with the reviewer and have tested if the mutation of the critical Y in the NCOA2 peptide to V would abrogate binding, and if mutation of the V in the NCOA1 & 3 peptides would enhance binding and found this to be the case (see previous comment). The alignment between NCOA1, 2 and 3 is shown in the new **Fig. S6**. Additionally, we tested the peptide containing LALLLL from N (SARS-CoV-2) and found it had slightly lower affinity compared to the NCOA2 peptide. We have added this information in the text and in **Figure S1**.

Comment

10- Mapping of the binding site: The NCOA2 and NYNRIN peptides bind to the same site as shown by the competitive displacement experiments. This site is proposed to match the binding site of the LALLLL motif based on structural predictions using Alphafold. While the AF model has reasonable quality, it is still a model and requires additional experimental support to draw a sound conclusion. The authors test displacement with full length N which contains the LALLLL motif plus an additional Ub1 binding site. The displacement experiment shows an increase in signal upon adding N, suggesting that a ternary complex is forming, and so the presence of the additional binding site in N complicates the analysis of the experiment. The claim that all peptides bind to the same cleft in Ub1 need additional support from one of two experiments: 1) showing that the isolated LALLLL motif can compete for binding of the NCOA2 motif to Ub1 and/or 2) performing a mutation on the proposed Ub1 binding cleft. Also, the structure of the LALLLL motif bound to Ub1 should be shown overlaid with the AF predictions for the peptides identified in this work.

Reply

We agree with the reviewer. We obtained the LALLL containing peptide, and found it to compete for binding with the NCOA2 motif, with an affinity that was 2-3 fold lower compared to NCOA2 peptide. Additionally, we created a pocket mutant of Nsp3 Ub1 and found that the mutations in the proposed binding pocket abrogated binding, confirming our modeling predictions. We have added these results to the text and to **Figure 3**.

We modified **Figure S4** to include the LALLLL motif bound to Ub1 overlaid with our AF2 predictions.

Comment

11- Line 219-222: “The peptide SPOT array showed a strong signal for binding of the MBOAT116-31 peptide to Nsp3 ADRP, suggesting that the central 21-HPLSE-25 residues are critical for binding (Figure 3C, residues in bold).” The description of the results is unclear: the SPOT array shows that mutation of each residue within the HPLSE motif strongly decrease binding, which indicates that this region is the main determinant for binding. This should be stated more clearly in the results text.

Reply

We have clarified this in the text.

Comment

12- Structural mapping of the binding site: If the predictions are low confidence (score ~30), they can't be used to make claims about the binding site in the absence of supporting mutagenesis of the ADRP binding

cleft and/or direct competition experiments with motifs known to bind to the proposed site. If no additional experiments are performed, showing the models in main text figures should be avoided and the conclusions of the paragraph below amended to indicate that while both peptides seem to bind to the same region in ADRP, no clear conclusions can be drawn regarding the binding site:

The ColabFold predictions of the Nsp3 ADRP binding peptides did not converge with high confidence (pLDDT of ~30; Figure S3). However, manual inspection indicated that the peptide binding was restricted to the surface between the N-terminal beta sheet (β 1) and the C-terminal alpha helix (α 6) (Figure 3B, Figure S3)”

Reply

We agree with the reviewer and mutated for each protein the predicted peptide binding pockets and determined the effect on the affinities of the peptides. For NSP3 Ubl1 and SUD-M the results nicely supported the predicted binding sites, while for Nsp3 ADRP and NSP9 the results did not support the predicted binding sites. In line with these results, we removed the Nsp3 ADRP prediction from the main figure.

Comment

Nsp3 SUD-M:

13- Same as point 12 for ADRP, the proposed binding site obtained with the AF prediction is of moderate quality and should be validated with mutagenesis of the SUD-M binding cleft. If no additional mapping experiments are performed, showing the AF models in main text figures should be avoided and the conclusions of the paragraph amended.

Reply

We agree and in order to map the binding site we expressed and purified pocket mutants of the domains. In the case of SUD-M the mutation of key residues in the predicted binding pocket abrogated binding corroborating the predicted binding site. We have added the information to the text and in **Figure 3E**.

Comment

14- Lines 241-245: The agreement between the relevant residues identified in the SPOT array and the face of the helix facing the domain, should be better explained in text and the residues labeled in the structure of SUD-M shown in Fig. 3B to facilitate the analysis.

Reply

In Figure 3B the issue with showing the interaction interface is that the view is obscured by the residues surrounding the binding site. We tried several different views that resulted in even poorer visibility of interacting residues. We have expanded the description in the figure legend to facilitate easier understanding and kindly request to leave the figure as is.

Comment

15- For the conclusions of the Nsp3 section, depending on the additional experiments performed, the writing should be improved to discriminate between what was learnt of the mapping the binding sites in the human proteins and determining the motif specificity, which is different from mapping (or not) of the binding site on the different domains, with the latter requiring additional support.

Reply

We have revised the text as detailed in the points above.

Comment

Nsp9

16- The set of peptides shown & tested in Fig. 4B are different from the set of peptides shown in Fig.2B. Please explain the selection criteria. Also, the authors should explain how the regular expression for the enriched motif was derived.

Reply

The selections of peptides have now been better explained. We also complemented the set in Fig. 2B to include the peptides used in the affinity measurements.

Comment

17- The authors adjust the motif pattern to GPhixPhi[GDP] based on the SPOT array performed with NKRF. However, this pattern is not supported by the SPOT array of LMTK3: Phe→Ala substitution at p2 yields no change in binding and Gly→Ala substitution at p5 strongly increases binding. The LMTK3 results need to be discussed and the conclusions amended. Unless additional mutagenesis experiments are performed, the pattern can only be claimed for NKRF, and a different binding specificity is observed in LMTK3 suggesting a broader binding pattern. Mutations following the identified motif increase binding for both peptides, and this result also needs to be discussed and interpreted.

Reply

We agree and have modified the text.

Comment

18- Nsp9 can exist as a monomer (i.e., when forming part of the Nsp7/8/9/12 RNA capping complex) and can form a homodimer stabilized by a dynamic C-terminal helix. The authors propose that since the peptide pattern resembles the Nsp9 C-terminal helix involved in dimerization, these peptides could bind to the dimerization interface and interfere with dimer formation. Since the AF predictions do not converge on a common binding site for different peptides, the authors perform NMR experiments to map the binding site on Nsp9 using three peptides. The first analysis is the chemical shift perturbation, which is mapped on the Nsp9 structure (Fig. 4C/E). The mapping of the binding site to the dimer interface is complicated by the fact that the C-terminal helix peaks are missing (as previously reported) and multiple other sites across Nsp9 loops and the hydrophobic core show changes in CS. The authors claim very similar CS perturbation patterns for the three peptides that support a common binding mode, but this would be more clearly evidenced by showing per-residue I/I₀ plots and CS plots for the three complexes. Please add this information to the supplementary info and possibly Main Figure: are other surface residues identified? This first set of experiments does not achieve unequivocal mapping of the binding interface.

Reply

Based on the comments we returned to the original experiments and reanalyzed our conclusions. Moreover, we obtained the Nsp9 construct where the C-terminal GxxxG motif is mutated to prevent dimerization. As this construct still bound to the NKRF peptide it was clear that the dimer interface is not the interface of the binding for the identified human ligands. We have re-written the results of the NMR experiments and backed our new conclusions with supplementary figures S9 and S11. We also provided chemical shift perturbation plot (**Figure 5E**) to support the notion that human peptide ligands perturb a similar pattern of amino acids on the surface and in the core of Nsp9.

Comment

19- Another piece of evidence is from analyzing whether addition of the peptides induces dissociation of the Nsp9 dimer. However, this is done by MW estimation from the T₁/T₂ relaxation times, which have a contribution from the molecule MW and shape and is therefore not unequivocal. Free Nsp9 has $t_c=15.3\text{ns}$ ($MW_{app} = 29\text{ KDa}$) and motif bound Nsp9 has $t_c=9.9\text{ns}$ ($MW_{app} = 18.8\text{ KDa}$). While the results suggest free Nsp9 is dimeric and bound Nsp9 is monomeric, there are other explanations (e.g., free Nsp9 could be

an extended monomer). A direct proof for the oligomeric state can be obtained by performing SEC-MALS or SEC-SLS measurements of free and NOTCH4-bound Nsp9, allowing an unequivocal determination of the MW of the species.

Reply

We performed SEC-MALS experiments but they were not conclusive. Therefore, we have amended our conclusions to avoid overinterpretation of our results.

Comment

20- To further test whether the peptides bound through the C-terminal helix the authors used a delta-Ct Nsp9 construct. This construct involves a 20-aa deletion that may disrupt Nsp9 folding and/or cause non-native oligomerization that confounds the result of the binding experiment. Therefore, biophysical characterization (Far-UV CD and SEC or SEC-SLS/MALS) should be performed to verify that the construct retains a native fold and secondary structure, and that it is a monomer. Without these controls, the experiment cannot be used to claim the peptides bind to the C-terminal helix region.

Reply

We did the Nsp9 pocket (binding site) mutant, and it did not abrogate the binding, thus showing that the GxxxG motif in Nsp9 is not the binding site of the peptides. Given that the deleted alpha helix ($\Delta\alpha$) Nsp9 variant does not bind peptides, but the glycine mutant does, suggests that the helix is involved in binding but that the GxxxG motif is not the core binding interface for the human peptides. To confirm that the Nsp9 $\Delta\alpha$ is folded we performed CD monitored temperature denaturation, which showed that the protein is folded at room temperature. We added the findings in the text and in **Figure S9**.

Additionally, since Nsp9 C-terminal helix double mutant bound FITC labeled peptide with same affinity as wt Nsp9 we amended our conclusion now stating that the interaction is not mediated through the C-terminal GxxxG motif.

Comment

21- Alternatively, several point mutations that stabilize the Nsp9 monomer state have been reported, and they could be used for mapping the proposed interaction site.

Reply

We did that and the results showed that the C-terminal helix GxxxG motif is not the binding interface of the peptides identified in this study. We have updated the text accordingly. See also the reply to the comment above.

Comment

Conclusions section:

22- Based on additional evidence obtained, the discussion should clearly review what information was gained for each domain-SLiM interaction. The claim that the novel human SLiMs bind to known binding clefts in the SARS-CoV-2 domains depend on additional experiments and should otherwise be amended.

Reply

We agree and have updated the text.

Comment

23- It would be useful to include a broader discussion on the possible role of the human SLiMs and the domains they target in human cells. Did the authors perform structural mapping studies to identify possible folds in human proteins that resemble the SARS-CoV-2 domains and could be targeted by these SLiMs? Are the human SLiMs conserved in the identified proteins?

Reply

As we don't know the human targets (if any) of the novel SLiMs it becomes challenging to discuss them in terms of possible roles. We evaluated the conservation of the motifs, and showed that some of the sequences within the human peptides are conserved across vertebrate or mammalian species as noted above (**Figure S6**). However, what they might bind to in terms of human proteins cannot be predicted accurately. Adding to the complexity, human and viral proteins may use completely different folds to bind to the same peptide sequences (PMID: 36504386). Thus, identifying the potential human binding partners of the peptides would require experimental efforts such as peptide-pulldowns coupled to mass-spectrometry. Such an endeavour would be a large study in itself.

Comment

Minor points:

- Fig. 2: all panels must be cited specifically in the main text (only figure and not figure panels are cited).

Reply

Done

Comment

- Fig. 3a legend says the data are presented as means \pm SD but the n is not defined. Kd values vary slightly with respect to Table S4, correct. Figure 3c, SPOT arrays define n value.

Reply

Done

Comment

- Figure 3B: Label the residues shown as sticks so they can be identified in the sequences of the motifs shown in Fig. 3c.

Reply

Done

Comment

- line 197/198: “This finding clarified the lack of binding of NCOA1 and NCOA2 paralogues” should say “This finding clarified the lack of binding of NCOA1 and NCOA3 paralogues”. However, this claim should be further supported by binding of individual motifs.

Reply

Fixed

Comment

- Line 475 Discussion “mass spectroscopy” is probably “mass spectrometry”

Reply

Corrected

Comment

- Line 480-481: “The Nsp9 dimer interface in the C-terminal helix with the GxxxG motif at its core has been proposed as a valid target for inhibitor development”. The sentence is unclear, rephrase.

Reply

Done

Comment

- Fig. S3 the line in the pIDDT plots probably represents the break between the domain and peptide in the prediction but this is not explained. Explain in figure legend.

Reply

Done

Comment

- Fig S4: if these are duplicate experiments explain. Provide raw SPOT array data.

Reply

SPOT arrays were performed as technical duplicates or triplicates as explained under previous points. All data are presented.

REVIEWERS' COMMENTS

Reviewer #1 (Remarks to the Author):

The authors addressed all of my comments, I would recommend publication.

Reviewer #2 (Remarks to the Author):

All my comments and criticisms have now been adequately addressed.

One more small thing to consider: When discussing the additional work included on mapping of the inhibitory effect of the Nsp9-binding NOTCH4 derived peptide, the authors could be more specific what they mean with "viral replication", and say e.g. RNA replication when appropriate.

Reviewer #3 (Remarks to the Author):

I am very pleased with the revision provided by the authors. They provide extensive revisions and additional work addressing all the points raised by myself and the other reviewers. The new experiments help clarify the molecular basis for many of the interactions and discard hypotheses that were not supported by the control/mapping experiments. Overall, the additional work has made the evidence in support of the conclusions much stronger. Besides very minor comments (see below) the work is suitable for publication.

1) It would be useful to cite the tool used to generate the logos in Figure 2 legend.

2) I appreciate that the authors removed peptide models with low confidence predictions. I recommend that in the figures where an AlphaFold2 model is shown for a peptide-domain interaction, the average pIDDT score of the peptide region that binds to the domain is reported in the figure legend, as this provides a measure of model confidence. Please also clarify if pIDDT scores reported are for the best model or for the average of five models.

REVIEWERS' COMMENTS

Reviewer #1 (Remarks to the Author):

The authors addressed all of my comments, I would recommend publication.

Reviewer #2 (Remarks to the Author):

All my comments and criticisms have now been adequately addressed.

One more small thing to consider: When discussing the additional work included on mapping of the inhibitory effect of the Nsp9-binding NOTCH4 derived peptide, the authors could be more specific what they mean with "viral replication", and say e.g. RNA replication when appropriate.

Reply: Yes, we have gone through the text and change viral replication to RNA replication in four places:

Results

Finally, we reasoned that if the NOTCH4-derived peptide act by targeting Nsp9 then it should block RNA replication. We therefore conducted a time of addition experiment where we added the inhibitor at distinct time points (1, 3, or 5 hours post infection; **Figure 7F**). The results showed that the inhibitor has the most potent effect 3 hours post infection, supporting the notion that it blocks RNA replication rather than interfering with viral entry or egress.

Legend to Fig 7F

(F) SARS CoV-2 replication in VeroE6 cells treated with 300 μ M NOTCH pen. or NOTCH pen. control peptides. RNA replication is presented as fold induction compared to input.

Discussion, second last sentence

We also showed that a subset of identified ligands inhibited RNA replication in cell culture, and that these peptides could be successfully converted into cell-penetrating anti-viral inhibitors.

Reviewer #3 (Remarks to the Author):

I am very pleased with the revision provided by the authors. They provide extensive revisions and additional work addressing all the points raised by myself and the other reviewers. The new experiments help clarify the molecular basis for many of the interactions and discard hypotheses that were not supported by the control/mapping experiments. Overall, the additional work has made the evidence in support of the conclusions much stronger. Besides very minor comments (see below) the work is suitable for publication.

1) It would be useful to cite the tool used to generate the logos in Figure 2 legend.

Reply: Done

2) I appreciate that the authors removed peptide models with low confidence predictions. I recommend that in the figures where an AlphaFold2 model is shown for a peptide-domain interaction, the average pIIDD score of the peptide region that binds to the domain is reported in the figure legend, as this provides a measure of model confidence. Please also clarify if pIIDD scores reported are for the best model or for the average of five models.

Reply: We have included the information in the main figures. Additionally, the pIIDD scores for all predictions are available in the Supplementary figures. The pIIDD scores are for the best model.